# When to Trust Your Simulator: Dynamics-Aware Hybrid Offline-and-Online Reinforcement Learning

**Haoyi Niu**[1*], **Shubham Sharma**[1,2*], **Yiwen Qiu**[1*], **Ming Li**[1,3*],
**Guyue Zhou**[1], **Jianming Hu**[1,4], **Xianyuan Zhan**[1,5†]

[1] Tsinghua University, Beijing, China
[2] Indian Institute of Technology, Bombay, India
[3] Shanghai Jiaotong University, Shanghai, China
[4] Beijing National Research Center for Information Science and Technology, Beijing, China
[5] Shanghai AI Laboratory, Shanghai, China

{t6.da.thu,shubh.am1107z,qywmei,liming18739796090}@gmail.com
hujm@mail.tsinghua.edu.cn
{zhouguyue,zhanxianyuan}@air.tsinghua.edu.cn

## Abstract

Learning effective reinforcement learning (RL) policies to solve real-world complex tasks can be quite challenging without a high-fidelity simulation environment. In most cases, we are only given imperfect simulators with simplified dynamics, which inevitably lead to severe sim-to-real gaps in RL policy learning. The recently emerged field of offline RL provides another possibility to learn policies directly from pre-collected historical data. However, to achieve reasonable performance, existing offline RL algorithms need impractically large offline data with sufficient state-action space coverage for training. This brings up a new question: is it possible to combine learning from limited real data in offline RL and unrestricted exploration through imperfect simulators in online RL to address the drawbacks of both approaches? In this study, we propose the Dynamics-Aware **H**ybrid **O**ffline-and-**O**nline Reinforcement Learning (H2O) framework to provide an affirmative answer to this question. H2O introduces a dynamics-aware policy evaluation scheme, which adaptively penalizes the Q-function learning on simulated state-action pairs with large dynamics gaps, while also simultaneously allowing learning from a fixed real-world dataset. Through extensive simulation and real-world tasks, as well as theoretical analysis, we demonstrate the superior performance of H2O against other cross-domain online and offline RL algorithms. H2O provides a brand new hybrid offline-and-online RL paradigm, which can potentially shed light on future RL algorithm design for solving practical real-world tasks.

## 1  Introduction

Over recent years, criticism against reinforcement learning (RL) continues to pour in regarding its poor real-world applicability. Although RL has demonstrated superhuman performance in solving complex tasks such as playing games [Mnih *et al.*, 2013; Silver *et al.*, 2017], its success is heavily dependent on availability of an unbiased interactive environment, either the real system or a high-fidelity simulator, as well as millions of unrestricted trials and errors. However, constructing high-fidelity simulators can be extremely expensive or even impossible due to complex system dynamics and unobservable information in the physical world. Less accurate simulators with simplifications, on the other hand, are easy to build, however, also lead to severe visual and dynamics sim-to-real gaps when learning

---

*Work done during internships at Institute for AI Industry Research (AIR), Tsinghua University.
†Corresponding author.

36th Conference on Neural Information Processing Systems (NeurIPS 2022).

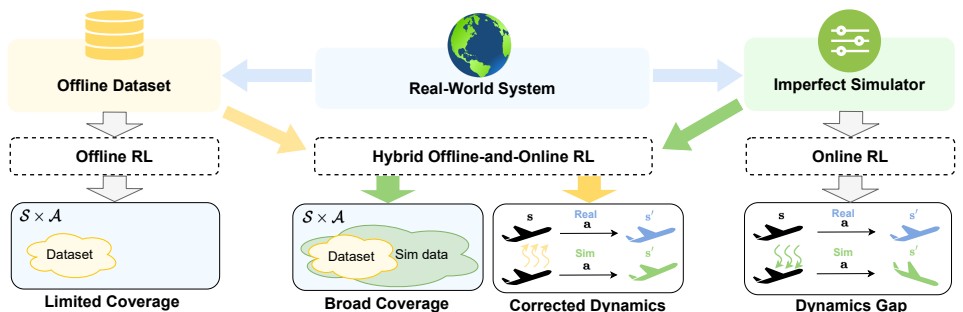

Figure 1: Conceptual illustration of the dynamics-aware hybrid offline-and-online RL framework

RL policies [Rao *et al.*, 2020; Peng *et al.*, 2018]. Among different sources of sim-to-real gaps, visual gaps are relatively well-addressed in a number of methods, e.g. through domain randomization [Tobin *et al.*, 2017], proper model setups that are less impacted by visual gaps [Lee *et al.*, 2021], or using a robust and transferable state-encoder [Bewley *et al.*, 2019; Rao *et al.*, 2020; Wang *et al.*, 2022]. Dynamics gaps on the other hand cause a systematic bias that is not easy to address through a modeling process. These dynamics gaps widely exist in almost every imperfect physical simulator with simplified dynamics, posing great challenges for RL when solving real-world tasks.

Recently, offline RL provides another possibility to directly learn policies from real-world data [Levine *et al.*, 2020; Fujimoto *et al.*, 2018; Kumar *et al.*, 2019], which has already been used in solving a number of practical problems, such as industrial control [Zhan *et al.*, 2022a], robotics [Lee *et al.*, 2021] and interactive recommendation [Xiao and Wang, 2021]. However, the policies learned through existing offline RL algorithms are often over-conservative due to the use of data-related policy constraints [Fujimoto *et al.*, 2019; Kumar *et al.*, 2019; Fujimoto and Gu, 2021], value regularizations [Kumar *et al.*, 2020; Kostrikov *et al.*, 2021; Xu *et al.*, 2022b], or in-sample learning [Kostrikov *et al.*, 2022; Xu *et al.*, 2022a] to combat the distributional shift [Kumar *et al.*, 2019]. This hurts the performance of offline RL policies and makes their performances strongly depend on the size and state-action space coverage of the offline dataset. In real-world scenarios, data collection can be expensive or restrictive. The state-action space coverage of actual offline datasets can be quite narrow, which directly limits the potential performance of offline RL policies.

Despite all the drawbacks, online RL with an imperfect simulator and offline RL with the real-world dataset bear indispensable potential when jointly used to learn high-performance policies and overcome the sim-to-real issues. Within a simulator, online RL agents can perform unrestricted exploration and have an access to a great quantity and diversity of state-action data, while offline datasets may have poor coverage. On the other hand, real-world datasets contain perfect dynamics information, which can be readily used to guide and correct erroneous information in online training with simulation data. This intuition has motivated researchers to explore how to incorporate offline real-world data with simulation-based online training, however, existing studies are only restricted to pure online [Eysenbach *et al.*, 2020] or offline [Liu *et al.*, 2022] learning settings. None of the prior works has successfully combined offline and online learning organically, while simultaneously handling the dynamics gaps between these two regimes.

In this paper, we present the first Hybrid Offline-and-Online Reinforcement Learning framework (H2O) as conceptually illustrated in Figure 1, which enables simultaneous policy learning with offline real-world datasets and simulation rollouts. The core of H2O is the introduction of dynamics-aware policy evaluation, which adaptively pushes down or pulls up Q-values on simulated data according to the dynamics gap evaluated against real data. We provide analyses and insights on why such dynamics-aware learning can potentially bridge the sim-to-real gap. Finally, we analyze the impact of each design component in H2O and demonstrate its superior performance against other cross-domain RL algorithms through extensive simulation and real-world experimental validations.

## 2 Related Work

### 2.1 Reinforcement Learning Using Simulators with Dynamics Gap

Dealing with dynamics gaps in simulators has long been recognized as a challenging task. The most straightforward approach is to calibrate the simulated dynamics with the real-world counterpart using

system identification methods [Ljung, 1998; Rajeswaran *et al.*, 2016; Chebotar *et al.*, 2019]. These methods typically need plenty of offline or even real-world interaction data to tune the parameters of the simulator [Yu *et al.*, 2017], which sometimes can be costly or offer little improvement if the simulator is over-simplified. Domain Randomization (DR) [Peng *et al.*, 2018; Andrychowicz *et al.*, 2020] provides another perspective that trains RL policies in a series of randomized simulated dynamics. DR has been shown to yield more adaptable policies in real-world, yet it often needs manually-specified randomized parameters and nuanced randomization distributions [Vuong *et al.*, 2019]. More recently, Dynamics Adaptation, e.g., DARC [Eysenbach *et al.*, 2020], augments the simulation reward $r(\mathbf{s}, \mathbf{a})$ with a dynamics gap-related penalization term $\Delta r(\mathbf{s}', \mathbf{s}, \mathbf{a})$ in online simulation-based RL training, derived from minimizing the divergence of distributions over real and simulation trajectories. DARC uses real data and trains two discriminators to evaluate $\Delta r(\mathbf{s}', \mathbf{s}, \mathbf{a})$, rather than directly leveraging the data for RL training. In a similar vein, DARA [Liu *et al.*, 2022] is the pure offline version of DARC, sacrificing the unlimited online exploration in the simulated domain. While existing RL studies have tackled the dynamics gap issue from different perspectives, none of them is able to develop a structured framework by combining offline data with online learning, simultaneously addressing the dynamics gap.

## 2.2 Offline Reinforcement Learning

The recently emerged offline RL methods provide a new alternative to learning policies directly from a fixed, pre-collected dataset. A key challenge of offline RL is the *distributional shift* problem [Kumar *et al.*, 2019; Levine *et al.*, 2020], which is caused by counterfactual queries of function approximators (e.g., value function, police network) on out-of-distribution (OOD) samples. Conventional off-policy RL methods will suffer severe value overestimation due to distributional shift. Existing offline RL methods solve this problem by introducing various data-related regularizations to constrain policy learning from deviating too much from the offline dataset, such as using data distribution, support or dataset distance based policy constraints [Fujimoto *et al.*, 2019; Kumar *et al.*, 2019; Wu *et al.*, 2019; Fujimoto and Gu, 2021; Xu *et al.*, 2021; Li *et al.*, 2022a], regularizing the value function on OOD data [Kumar *et al.*, 2020; Kostrikov *et al.*, 2021; Yu *et al.*, 2021; Li *et al.*, 2022b; Xu *et al.*, 2022b], in-sample learning [Kostrikov *et al.*, 2022; Xu *et al.*, 2022a], and adding uncertainty-based penalties [Yu *et al.*, 2020; Kidambi *et al.*, 2020; Zhan *et al.*, 2022a,b]. Conservatism and pessimism are central principles [Buckman *et al.*, 2020] used in most existing offline RL algorithms, which cause the performances of these methods to heavily depend on the size, quality, and state-action space coverage of the offline dataset. In most practical settings, getting a dataset with large coverage is impractical. Meanwhile, if online interaction with a simulator is involved, although may not be accurate, it can substantially supplement the state-action space coverage of offline datasets in policy learning. This motivates a new paradigm of combining both offline and online RL, which is explored in our proposed H2O framework.

# 3 Background

## 3.1 Reinforcement Learning

We consider the standard Markov Decision Process (MDP) setting, which is specified by a tuple $\mathcal{M} := (\mathcal{S}, \mathcal{A}, r, P_{\mathcal{M}}, \rho, \gamma)$, where $\mathcal{S}$ and $\mathcal{A}$ are the state and action spaces, $r(\mathbf{s}, \mathbf{a})$ is the reward function, $P_{\mathcal{M}}(\mathbf{s}'|\mathbf{s}, \mathbf{a})$ refers to the transitional dynamics under $\mathcal{M}$, $\rho(\mathbf{s})$ depicts the initial state distribution, and $\gamma \in (0, 1)$ is the discount factor. The goal of RL is to learn a policy $\pi(\mathbf{a}|\mathbf{s})$ that maps a state $\mathbf{s}$ to an action $\mathbf{a}$ so as to maximize the expected cumulative discounted reward starting from $\rho$, i.e., $J(\mathcal{M}, \pi) = \mathbb{E}_{\mathbf{s}_0 \in \rho, \mathbf{a}_t \sim \pi(\cdot|\mathbf{s}_t), \mathbf{s}_{t+1} \sim P_{\mathcal{M}}(\cdot|\mathbf{s}, \mathbf{a})} \left[\sum_{t=0}^{\infty} \gamma^t r(\mathbf{s}_t, \mathbf{a}_t)\right]$. In conventional actor-critic formalism [Barto *et al.*, 1983; Sutton *et al.*, 1998], one learns an approximated Q-function $\hat{Q}$ by minimizing the squared Bellman error (referred as *policy evaluation*), and optimizes the policy $\pi$ by maximizing the Q-function (referred as *policy improvement*) as follows:

$$\hat{Q} \leftarrow \arg\min_{Q} \mathbb{E}_{\mathbf{s}, \mathbf{a}, \mathbf{s}' \sim \mathcal{U}} \left[ \left( Q(\mathbf{s}, \mathbf{a}) - \hat{\mathcal{B}}^{\pi} \hat{Q}(\mathbf{s}, \mathbf{a}) \right)^2 \right] \qquad \text{(Policy Evaluation)} \qquad (1)$$

$$\hat{\pi} \leftarrow \arg\max_{\pi} \mathbb{E}_{\mathbf{s}, \mathbf{a} \sim \mathcal{U}} \left[ \hat{Q}(\mathbf{s}, \mathbf{a}) \right] \qquad \text{(Policy Improvement)} \qquad (2)$$

where $\mathcal{U}$ can either be the replay buffer $B$ generated by a previous version of policy $\hat{\pi}$ through online environment interactions, or can be a fixed dataset $\mathcal{D} = \left\{ \left( \mathbf{s}_t^i, \mathbf{a}_t^i, \mathbf{s}_{t+1}^i, r_t^i \right) \right\}_{i=1}^{n}$ as in offline

RL setting. $\hat{\mathcal{B}}^\pi$ is the Bellman operator, which is often used as the Bellman evaluation operator $\hat{\mathcal{B}}^\pi \hat{Q}(\mathbf{s}, \mathbf{a}) = r(\mathbf{s}, \mathbf{a}) + \gamma \mathbb{E}_{\mathbf{a}' \sim \hat{\pi}(\mathbf{a}'|\mathbf{s}')} \left[ \hat{Q}(\mathbf{s}', \mathbf{a}') \right]$.

## 3.2 Offline RL via Value Regularization

In the offline setting, performing the standard Bellman update in Eq. (1) can result in serious overestimation over Q-values due to distributional shift. An effective approach to alleviating this problem is to regularize the value function on OOD actions, which is introduced in CQL [Kumar *et al.*, 2020]. CQL learns a policy $\hat{\pi}$ upon a lower-bounded Q-function that additionally pushes down the Q-values on actions induced by $\mu(\mathbf{a}|\mathbf{s})$ and pulls up Q-values on trustworthy offline data:

$$\min_Q \max_\mu \alpha \left( \mathbb{E}_{\mathbf{s} \sim \mathcal{D}, \mathbf{a} \sim \mu(\cdot|\mathbf{s})}[Q(\mathbf{s}, \mathbf{a})] - \mathbb{E}_{\mathbf{s}, \mathbf{a} \sim \mathcal{D}}[Q(\mathbf{s}, \mathbf{a})] \right) + \mathcal{E}\left( Q, \hat{\mathcal{B}}^\pi \hat{Q} \right) \tag{3}$$

where the dataset $\mathcal{D}$ is collected by some behavioral policy $\pi_\mathcal{D}(\cdot|\mathbf{s})$, and $\mu(\cdot|\mathbf{s})$ is some sampling distribution, which is often taken as $\hat{\pi}(\cdot|\mathbf{s})$ in prior works [Yu *et al.*, 2021; Li *et al.*, 2022b]. $\mathcal{E}\left( Q, \hat{\mathcal{B}}^\pi \hat{Q} \right)$ denotes the Bellman error in Eq. (1). It is worth noting that the above value regularization framework has the potential to handle data from two sources with some preferences, i.e., unreliable pushed-down data and reliable pulled-up data. This motivates us to devise the H2O framework to perform simultaneous offline and online policy learning using a similar value regularization recipe.

# 4 Hybrid Offline-and-Online Reinforcement Learning

Naïvely combining offline and online data from different distributions may cause the mismatch between data distributions, giving rise to issues similar to the distributional shift problem as in offline RL settings [Kumar *et al.*, 2019]. Caution needs to be taken when dealing with trustworthy offline data from the real world and the potentially problematic simulated online rollouts (described in Section 4.1). Secondly, the dynamics model underpinning the simulator is only roughly aligned with the real-world counterpart, and the sim-to-real dynamics gaps can be heterogeneous across different state-action pairs. Hence uniform value regularization upon all simulated samples may not be appropriate to fully leverage the potential of an imperfect simulator. A well-founded dynamics gap measurement for adaptive value regularization is needed (Section 4.2). Lastly, it is also worth noting that the biased simulated dynamics could induce erroneous next-state predictions, which may lead to problematic Bellman updates. Such errors also need to be properly considered in the algorithm design (Section 4.3). We approach all the aforementioned issues with what we call Dynamics-Aware Hybrid Offline-and-Online (H2O) RL, which is described in the following content.

## 4.1 Incorporating Offline Data in Online Learning

To address the potential mismatch of training data distributions from different data sources, existing online RL frameworks are not suitable, however, offline RL approaches such as value regularization can provide a viable foundation. We modify the previous value regularization scheme in offline RL and propose the dynamics-aware policy evaluation, which has the following general form:

$$\min_Q \max_{d^\phi} \beta \left[ \mathbb{E}_{\mathbf{s}, \mathbf{a} \sim d^\phi(\mathbf{s}, \mathbf{a})}[Q(\mathbf{s}, \mathbf{a})] - \mathbb{E}_{\mathbf{s}, \mathbf{a} \sim \mathcal{D}}[Q(\mathbf{s}, \mathbf{a})] + \mathcal{R}(d^\phi) \right] + \widetilde{\mathcal{E}}\left( Q, \hat{\mathcal{B}}^\pi \hat{Q} \right) \tag{4}$$

where $\beta$ is a positive scaling parameter. $d^\phi(\mathbf{s}, \mathbf{a})$ is a particular state-action sampling distribution that associates with high dynamics-gap samples, and $\mathcal{R}(d^\phi)$ is a regularization term for $d^\phi$ to enforce this designed behavior. Ideally, we would want to penalize Q-values at the simulated samples with high dynamics gaps, rather than all of the samples. $\widetilde{\mathcal{E}}(Q, \hat{\mathcal{B}}^\pi \hat{Q})$ denotes the modified Bellman error of the mixed data from offline dataset $\mathcal{D}$ and the simulation rollout samples in online replay buffer $B$, which are generated by the real MDP $\mathcal{M}$ and the simulated MDP $\widehat{\mathcal{M}}$ respectively. Specifically, the Bellman error of the simulated data needs to be fixed. The terms marked in red and blue are the key design elements that account for the dynamics gaps in offline-and-online RL policy learning, which will be discussed in the following sub-sections.

## 4.2 Adaptive Value Regularization on High Dynamics-Gap Samples

To achieve adaptive value regularization, we can use the minimization term $\mathbb{E}_{\mathbf{s}, \mathbf{a} \sim d^\phi(\mathbf{s}, \mathbf{a})}[Q(\mathbf{s}, \mathbf{a})]$ to penalize the high-dynamics gap simulation samples, and use the maximization term $\mathbb{E}_{\mathbf{s}, \mathbf{a} \sim \mathcal{D}}[Q(\mathbf{s}, \mathbf{a})]$ to cancel the penalization on real offline data. The key question is how to properly design $d^\phi(\mathbf{s}, \mathbf{a})$ to

assign high probabilities to high-dynamics gap samples. In our study, we leverage the regularization term $\mathcal{R}(d^\phi)$ to control the behavior of $d^\phi(\mathbf{s}, \mathbf{a})$. We choose $\mathcal{R}(d^\phi) = -D_{KL}(d^\phi(\mathbf{s}, \mathbf{a})\|\omega(\mathbf{s}, \mathbf{a}))$, where $D_{KL}(\cdot\|\cdot)$ is the Kullback-Leibler (KL) divergence and $\omega(\mathbf{s}, \mathbf{a})$ is a distribution that characterizes the dynamics gaps for samples in the state-action space. Hence maximizing over $\mathcal{R}(d^\phi)$ draws closer between $d^\phi$ and $\omega$, and achieves our desired behavior. Note that the inner maximization problem over $d^\phi$ in Eq. (4) under this design corresponds to the following optimization problem:

$$\max_{d^\phi} \mathbb{E}_{\mathbf{s}, \mathbf{a}\sim d^\phi(\mathbf{s}, \mathbf{a})}[Q(\mathbf{s}, \mathbf{a})] - D_{KL}(d^\phi(\mathbf{s}, \mathbf{a})\|\omega(\mathbf{s}, \mathbf{a})) \quad \text{s.t.} \sum_{\mathbf{s}, \mathbf{a}} d^\phi(\mathbf{s}, \mathbf{a}) = 1, d^\phi(\mathbf{s}, \mathbf{a}) \geq 0 \quad (5)$$

Above optimization problem admits a closed-form solution $d^\phi(\mathbf{s}, \mathbf{a}) \propto \omega(\mathbf{s}, \mathbf{a}) \exp(Q(\mathbf{s}, \mathbf{a}))$ (see the derivation in Appendix A.1). Plugging this back into Eq. (4), the first term now corresponds to a weighted soft-maximum of Q-values at any state-action pair and the original problem transforms into the following form:

$$\min_{Q} \beta \left( \log \sum_{\mathbf{s}, \mathbf{a}} \omega(\mathbf{s}, \mathbf{a}) \exp(Q(\mathbf{s}, \mathbf{a})) - \mathbb{E}_{\mathbf{s}, \mathbf{a}\sim\mathcal{D}}[Q(\mathbf{s}, \mathbf{a})] \right) + \widetilde{\mathcal{E}}\left(Q, \hat{\mathcal{B}}^\pi \hat{Q}\right) \quad (6)$$

This result is intuitively reasonable, as we are penalizing more on Q-values with larger $\omega(\mathbf{s}, \mathbf{a})$, corresponding to those high dynamics-gap simulation samples. Then the next question is, how do we practically evaluate $\omega(\mathbf{s}, \mathbf{a})$ given the offline dataset and the online simulation rollouts? Specifically, we can measure the dynamics gap between real and simulated dynamics on a state-action pair $(\mathbf{s}, \mathbf{a})$ as $u(\mathbf{s}, \mathbf{a}) := D_{KL}(P_{\widehat{\mathcal{M}}}(\mathbf{s}'|\mathbf{s}, \mathbf{a})\|P_{\mathcal{M}}(\mathbf{s}'|\mathbf{s}, \mathbf{a})) = \mathbb{E}_{\mathbf{s}'\sim P_{\widehat{\mathcal{M}}}} \log(P_{\widehat{\mathcal{M}}}(\mathbf{s}'|\mathbf{s}, \mathbf{a})/P_{\mathcal{M}}(\mathbf{s}'|\mathbf{s}, \mathbf{a}))$. $\omega(s, a)$ can thus be represented as a normalized distribution of $u$, i.e., $\omega(\mathbf{s}, \mathbf{a}) = u(\mathbf{s}, \mathbf{a})/\sum_{\widetilde{\mathbf{s}}, \widetilde{\mathbf{a}}} u(\widetilde{\mathbf{s}}, \widetilde{\mathbf{a}})$. Now, the challenge is how to evaluate the dynamics ratio $P_{\widehat{\mathcal{M}}}/P_{\mathcal{M}}$. Note that according to Bayes' rule:

$$\begin{aligned}\frac{P_{\widehat{\mathcal{M}}}(\mathbf{s}'|\mathbf{s}, \mathbf{a})}{P_{\mathcal{M}}(\mathbf{s}'|\mathbf{s}, \mathbf{a})} &= \frac{p(\mathbf{s}'|\mathbf{s}, \mathbf{a}, \text{sim})}{p(\mathbf{s}'|\mathbf{s}, \mathbf{a}, \text{real})} = \frac{p(\text{sim}|\mathbf{s}, \mathbf{a}, \mathbf{s}')}{p(\text{sim}|\mathbf{s}, \mathbf{a})} / \frac{p(\text{real}|\mathbf{s}, \mathbf{a}, \mathbf{s}')}{p(\text{real}|\mathbf{s}, \mathbf{a})} \\ &= \frac{p(\text{sim}|\mathbf{s}, \mathbf{a}, \mathbf{s}')}{p(\text{real}|\mathbf{s}, \mathbf{a}, \mathbf{s}')} / \frac{p(\text{sim}|\mathbf{s}, \mathbf{a})}{p(\text{real}|\mathbf{s}, \mathbf{a})} = \frac{1 - p(\text{real}|\mathbf{s}, \mathbf{a}, \mathbf{s}')}{p(\text{real}|\mathbf{s}, \mathbf{a}, \mathbf{s}')} / \frac{1 - p(\text{real}|\mathbf{s}, \mathbf{a})}{p(\text{real}|\mathbf{s}, \mathbf{a})} \end{aligned} \quad (7)$$

Hence, we can approximate $p(\text{real}|\mathbf{s}, \mathbf{a}, \mathbf{s}')$ and $p(\text{real}|\mathbf{s}, \mathbf{a})$ with a pair of discriminators $D_{\Phi_{sas}}(\cdot|\mathbf{s}, \mathbf{a}, \mathbf{s}')$ and $D_{\Phi_{sa}}(\cdot|\mathbf{s}, \mathbf{a})$ respectively that are optimized with standard cross-entropy loss between real offline data and the simulated samples as in DARC [Eysenbach *et al.*, 2020].

### 4.3 Fixing Bellman Error due to Dynamics Gap

Due to the existence of dynamics gaps in the simulator, directly computing the Bellman error for simulated samples in the online replay buffer $B$ (i.e., $\mathbb{E}_{\mathbf{s}, \mathbf{a}, \mathbf{s}'\sim B}[(Q - \hat{\mathcal{B}}^\pi \hat{Q})(\mathbf{s}, \mathbf{a})]^2)$ is problematic. This is because the next state $\mathbf{s}'$ comes from the potentially biased simulated dynamics $P_{\widehat{\mathcal{M}}}$, which can result in the miscalculation of target Q-values in Bellman updates. Ideally, we wish the next state $\mathbf{s}'$ comes from the real dynamics $P_{\mathcal{M}}$, however, such an $\mathbf{s}'$ given an arbitrary state-action pair $(\mathbf{s}, \mathbf{a}) \sim B$ is not obtainable during training. To fix this issue, we reuse the previously evaluated dynamics ratio in Eq. (7) as an importance sampling weight, and introduce the following modified Bellman error formulation on both offline dataset $\mathcal{D}$ and simulated data $B$:

$$\begin{aligned}\widetilde{\mathcal{E}}\left(Q, \hat{\mathcal{B}}^\pi \hat{Q}\right) &= \frac{1}{2}\mathbb{E}_{\mathbf{s}, \mathbf{a}, \mathbf{s}'\sim\mathcal{D}}\left[\left(Q - \hat{\mathcal{B}}^\pi \hat{Q}\right)(\mathbf{s}, \mathbf{a})\right]^2 + \frac{1}{2}\mathbb{E}_{\mathbf{s}, \mathbf{a}\sim B}\mathbb{E}_{\mathbf{s}'\sim P_{\mathcal{M}}}\left[\left(Q - \hat{\mathcal{B}}^\pi \hat{Q}\right)(\mathbf{s}, \mathbf{a})\right]^2 \\ &= \frac{1}{2}\mathbb{E}_{\mathbf{s}, \mathbf{a}, \mathbf{s}'\sim\mathcal{D}}\left[\left(Q - \hat{\mathcal{B}}^\pi \hat{Q}\right)(\mathbf{s}, \mathbf{a})\right]^2 + \frac{1}{2}\mathbb{E}_{\mathbf{s}, \mathbf{a}, \mathbf{s}'\sim B}\left[\frac{P_{\mathcal{M}}(\mathbf{s}'|\mathbf{s}, \mathbf{a})}{P_{\widehat{\mathcal{M}}}(\mathbf{s}'|\mathbf{s}, \mathbf{a})}\left(Q - \hat{\mathcal{B}}^\pi \hat{Q}\right)(\mathbf{s}, \mathbf{a})\right]^2 \end{aligned} \quad (8)$$

### 4.4 Practical Implementation

Combining Eq. (6) and (8), we obtain the final dynamics-aware policy evaluation procedure for H2O, which enables adaptive value regularization on high dynamics-gap samples as well as more reliable Bellman updates. H2O can be instantiated upon common actor-critic algorithms (e.g., Soft Actor-Critic (SAC) [Haarnoja *et al.*, 2018]). The pseudocode of H2O built upon SAC is presented in Algorithm 1, which uses the policy improvement objective from SAC and $\lambda$ is a temperature parameter auto-tuned during training. Other policy improvement objectives, or simply maximizing the expected Q-values as in Eq. (2) are also compatible with the H2O framework.

**Algorithm 1:** Dynamics-Aware Hybrid Offline-and-Online Reinforcement Learning (H2O)

---

**Data:** an offline dataset $\mathcal{D}$ from the real world, an imperfect simulator with biased dynamics $\widehat{\mathcal{M}}$

1 **Initialize:** critic network $Q_\theta$, target network $Q_{\bar\theta}$, actor network $\pi_\phi$, replay buffer $B = \varnothing$ for simulated transitions, discriminators $D_{\Phi_{sas}}(\cdot|\mathbf{s}, \mathbf{a}, \mathbf{s}')$ and $D_{\Phi_{sa}}(\cdot|\mathbf{s}, \mathbf{a})$

2 **for** *step* $t = 1, \cdots, T$ **do**

3     $B \leftarrow B \cup \text{ROLLOUT}(\pi_\phi, \widehat{\mathcal{M}})$;                        ▷ Collect simulated data

4     $\Phi_{sas} \leftarrow \Phi_{sas} - \eta \nabla_{\Phi_{sas}} \text{CROSSENTROPYLOSS}(B, \mathcal{D}, \Phi_{sas})$;     ▷ Update $D_{\Phi_{sas}}$

5     $\Phi_{sa} \leftarrow \Phi_{sa} - \eta \nabla_{\Phi_{sa}} \text{CROSSENTROPYLOSS}(B, \mathcal{D}, \Phi_{sa})$;       ▷ Update $D_{\Phi_{sa}}$

6     $\theta \leftarrow \theta - \eta_Q \nabla_\theta \left[ \beta \left( \log \sum \omega(\mathbf{s}, \mathbf{a}) \exp(Q_\theta(\mathbf{s}, \mathbf{a})) - \mathbb{E}_{\mathbf{s}, \mathbf{a} \sim \mathcal{D}} [Q_\theta(\mathbf{s}, \mathbf{a})] \right) + \widetilde{\mathcal{E}} \left( Q_\theta, \hat{\mathcal{B}}^\pi Q_{\bar\theta} \right) \right]$;

7     $\phi \leftarrow \phi + \eta_\pi \nabla_\phi \left[ \mathbb{E}_{\mathbf{s}, \mathbf{a} \sim \{\mathcal{D} \cup B\}} [Q_\theta(\mathbf{s}, \mathbf{a}) - \lambda \log \pi_\phi(\mathbf{a}|\mathbf{s})] \right]$;

8     **if** $t$ % target_update_period = 0 **then**

9        $\bar\theta \leftarrow (1 - \tau)\bar\theta + \tau\theta$;                     ▷ Soft update periodically

10     **end**

11 **end**

---

For practical considerations, we introduce two relaxations in our implementation. First, in Eq. (6), we need to sample from the whole state-action space to compute the weighted average of exponentiated Q-values, which can be impractical and unnecessary since not all state-action pairs are of interest for policy learning. Instead, we approximate this value using the state-action samples in the mini-batch of the simulated replay buffer $B$. Second, the evaluation of the KL divergence in $u(\mathbf{s}, \mathbf{a}) = \mathbb{E}_{\mathbf{s}' \sim P_{\widehat{\mathcal{M}}}} \log(P_{\widehat{\mathcal{M}}}(\mathbf{s}'|\mathbf{s}, \mathbf{a})/P_{\mathcal{M}}(\mathbf{s}'|\mathbf{s}, \mathbf{a}))$ involves sampling next states $\mathbf{s}'$ from $P_{\widehat{\mathcal{M}}}(\cdot|\mathbf{s}, \mathbf{a})$, which can be infeasible given a black-box simulator. We take a simplified approach that approximates the expected value by averaging over $N$ random samples from a Gaussian distribution $\mathcal{N}(\mathbf{s}', \hat\Sigma_{\mathcal{D}})$, where $\hat\Sigma_{\mathcal{D}}$ is the covariance matrix of states computed from the real offline dataset. Although this treatment is simple, we find it highly effective and produces good performance in empirical experiments. Please turn to Appendix B.1 for more implementation details.

## 5 Interpretation of Dynamics-Aware Policy Evaluation

In this section, we analyze H2O and provide intuition on how it works when combining both offline and online learning. Interestingly, we find that the final form of dynamics-aware policy evaluation as given in Eq. (6) and (8) actually leads to an equivalent adaptive reward adjustment term on the Q-function, which depends on the dynamics gap distribution $\omega(\mathbf{s}, \mathbf{a})$ and the state-action distribution of the dataset $d_{\mathcal{M}}^{\pi_{\mathcal{D}}}(\mathbf{s}, \mathbf{a})$. To show this, note that the weighted log-sum-exp term $\log \sum_{\mathbf{s}, \mathbf{a}} \omega(\mathbf{s}, \mathbf{a}) \exp(Q(\mathbf{s}, \mathbf{a})) = \log \mathbb{E}_{\mathbf{s}, \mathbf{a} \sim \omega(\mathbf{s}, \mathbf{a})} \exp(Q(\mathbf{s}, \mathbf{a}))$ in Eq. (6) can be bounded as follows (see Appendix A.2 for detailed proof):

$$\mathbb{E}_{\mathbf{s}, \mathbf{a} \sim \omega(\mathbf{s}, \mathbf{a})}[Q(\mathbf{s}, \mathbf{a})] \leq \log \mathbb{E}_{\mathbf{s}, \mathbf{a} \sim \omega(\mathbf{s}, \mathbf{a})} \exp(Q(\mathbf{s}, \mathbf{a})) \leq \mathbb{E}_{\mathbf{s}, \mathbf{a} \sim \omega(\mathbf{s}, \mathbf{a})}[Q(\mathbf{s}, \mathbf{a})] + \frac{\text{Var}_\omega[\exp(Q(\mathbf{s}, \mathbf{a}))]}{2 \exp(2Q_{min})} \quad (9)$$

where we denote the range of learned Q-values as $[Q_{min}, Q_{max}]$, and $\text{Var}_\omega[\exp(Q(\mathbf{s}, \mathbf{a}))]$ denotes the variance of $\exp(Q(\mathbf{s}, \mathbf{a}))$ under samples drawn from distribution $\omega(\mathbf{s}, \mathbf{a})$. The LHS inequality is a result of Jensen's inequality, and the RHS inequality is a direct result from [Liao and Berg, 2018]. Above inequalities suggest that $\mathbb{E}_{\mathbf{s}, \mathbf{a} \sim \omega(\mathbf{s}, \mathbf{a})}[Q(\mathbf{s}, \mathbf{a})]$ can be a reasonable approximation of our weighted log-sum-exp term if the Q-function takes large values and the gap between $\exp(Q_{min})$ and $\sqrt{\text{Var}_\omega[\exp(Q(\mathbf{s}, \mathbf{a}))]}$ is not too large. This can be practically satisfied if we let $\gamma \to 1$ and design the range of reward function as well as the episode done conditions properly.

With this approximation, we can consider the following approximated policy evaluation objective based on Eq. (6) and (8), which offers a much cleaner form for analysis:

$$\min_Q \beta \left( \mathbb{E}_{\mathbf{s}, \mathbf{a} \sim \omega(\mathbf{s}, \mathbf{a})}[Q(\mathbf{s}, \mathbf{a})] - \mathbb{E}_{\mathbf{s}, \mathbf{a} \sim \mathcal{D}}[Q(\mathbf{s}, \mathbf{a})] \right) + \widetilde{\mathcal{E}} \left( Q, \hat{\mathcal{B}}^\pi \hat{Q}^k \right) \quad (10)$$

The ablation study in Appendix C.1 manifests that this approximated policy evaluation objective only yields mild performance drops against the original version in most cases, making it a reasonable substitution for theoretical analysis. Assuming that the Q-function is tabular, we can find the Q-value

corresponding to the new objective by following an approximate dynamic programming approach and differentiating Eq. (10) with respect to $Q^k$ in iteration $k$ (see Appendix A.2 for details):

$$\hat{Q}^{k+1}(\mathbf{s}, \mathbf{a}) = (\hat{\mathcal{B}}^\pi \hat{Q}^k)(\mathbf{s}, \mathbf{a}) - \beta \left[ \frac{\omega(\mathbf{s}, \mathbf{a}) - d_{\mathcal{M}}^{\pi_{\mathcal{D}}}(\mathbf{s}, \mathbf{a})}{d_{\mathcal{M}}^{\pi_{\mathcal{D}}}(\mathbf{s}, \mathbf{a}) + d_{\widehat{\mathcal{M}}}^{\pi}(\mathbf{s}, \mathbf{a})} \right] \tag{11}$$

where $d_{\mathcal{M}}^{\pi_{\mathcal{D}}}(\mathbf{s}, \mathbf{a})$ and $d_{\widehat{\mathcal{M}}}^{\pi}(\mathbf{s}, \mathbf{a})$ are state-action marginal distributions under behavioral policy $\pi_{\mathcal{D}}$ and the learned policy $\pi$ respectively. If we represent the second term in the RHS of above equation as $\nu(\mathbf{s}, \mathbf{a})$, we can see that $\nu(\mathbf{s}, \mathbf{a})$ can be perceived as an adaptive reward adjustment term, which penalizes or boosts the reward at a state-action pair $(\mathbf{s}, \mathbf{a})$ according to the relative difference between dynamics gap distribution $\omega(\mathbf{s}, \mathbf{a})$ and marginal state-action distribution of the offline dataset $d_{\mathcal{M}}^{\pi_{\mathcal{D}}}(\mathbf{s}, \mathbf{a})$. Specifically, we can make the following interesting observations:

- If $\omega(\mathbf{s}, \mathbf{a}) > d_{\mathcal{M}}^{\pi_{\mathcal{D}}}(\mathbf{s}, \mathbf{a})$, then $\nu(\mathbf{s}, \mathbf{a})$ serves as a reward penalty. This corresponds to the case that either the state-action pair $(\mathbf{s}, \mathbf{a})$ has large dynamics gap, or it belongs to OOD or relatively low data density areas ($d_{\mathcal{M}}^{\pi_{\mathcal{D}}}(\mathbf{s}, \mathbf{a}) = 0$ or $< \omega(\mathbf{s}, \mathbf{a})$). Under such cases, we push down the Q-values and adopt conservatism similar to the typical treatment in offline RL settings [Kumar *et al.*, 2020]. The higher the dynamics gap is (larger $\omega$), the more we penalize the Q-function.

- If $\omega(\mathbf{s}, \mathbf{a}) < d_{\mathcal{M}}^{\pi_{\mathcal{D}}}(\mathbf{s}, \mathbf{a})$, then we are evaluating at some good data areas with low dynamics gap or have more information from offline data to control the potential dynamics gap in the simulated online samples. Under such cases, $\nu(\mathbf{s}, \mathbf{a})$ serves as a reward boost term and adopts optimism. Although it overestimates the Q-values, we argue that this can be beneficial. As it can promote exploration in these "good" data regions and potentially help reduce variance during training.

With the adaptive reward adjustment term $\nu$ defined in Eq. (11), we can further show that the value function $V(\mathbf{s})$ is underestimated at high dynamics gap areas, which allows for safe policy learning in our offline-and-online setting. Detailed theoretical analysis can be found in Appendix A.3.

## 6 Experiments

In this section, we present empirical evaluations of H2O. We start by describing our experimental environment setups and the cross-domain RL baselines for comparison. We then evaluate H2O against the baseline methods in simulation environments and on a real wheel-legged robot. Ablation studies and empirical analyses of H2O are also reported. Additional experiment settings, ablations, and results can be found in Appendix B, C, and D respectively.

### 6.1 Experimental Environment Setups

**Simulation-based experiments.**   We conduct simulation-based experiments in the MuJoCo physics simulator [Todorov *et al.*, 2012]. In particular, we construct three new simulation task environments (serve as the simulated environments) with intentionally introduced dynamics gaps upon the original MuJoCo-HalfCheetah task environments (serve as the real environments) by modifying the dynamics parameters: (1) **Gravity**: applying 2 times the gravitational acceleration in the simulation dynamics; (2) **Friction**: using 0.3 times the friction coefficient to make the agent harder to maintain balance; (3) **Joint Noise**: adding a random noise sampled from $\mathcal{N}(0, 1)$ on every dimension of the action space, mimicking a system with large control noise. As for the offline dataset from the real world (original simulation environment), we use the datasets of the corresponding task from standard offline RL benchmark D4RL [Fu *et al.*, 2020]. Specifically, we consider the Medium, Medium Replay and Medium Expert datasets, as in typical real-world scenarios, we do not use a random policy or do not have an expert policy for system control. The online training is performed on the modified simulation environment, and we evaluate the performance of the learned policy in the original unchanged MuJoCo environment in terms of the average return.

**Real-world experiments.**   We also evaluate the performance of H2O on a real wheel-legged robot that moves on a pair of wheels to keep it balanced, as illustrated in Figure 2(a). The state space of the robot is a quadratic-tuple $\mathcal{S} = (\theta, \dot{\theta}, x, \dot{x})$, where $\theta$ denotes the forward tilt angle of the body, $x$ is the displacement of the robot, $\dot{\theta}$ and $\dot{x}$ are the angular and linear velocity respectively. The control action is the torque $\tau$ of the motors at the two wheels. Here we construct two tasks for real-world validation: (1) **Standing still**: We want to make the robot stand still and not move or fall down; (2) **Moving straight**: We want to control the robot to move forward at a target velocity $v$ while keeping its balance. We record 100,000 human-controlled transitions with different reward functions for

Table 1: Average returns for MuJoCo-HalfCheetah tasks. Results are averaged over 5 random seeds.

| Dataset | Unreal dynamics | SAC(sim) | CQL(real) | DARC | DARC+ | H2O |
|---|---|---|---|---|---|---|
| Medium | Gravity | 4513±513 | 6066±73 | 5011±456 | 5706±440 | **7085±416** |
| | Friction | 2684±2646 | 6066±73 | 6113±104 | 6047±112 | **6848±445** |
| | Joint Noise | 4137±805 | 6066±73 | 5484±171 | 5314±520 | **7212±236** |
| Medium Replay | Gravity | 4513±513 | 5774±214 | 5105±460 | 4958±540 | **6813±289** |
| | Friction | 2684±2646 | 5774±214 | 5503±263 | 5288±100 | **5928±896** |
| | Joint Noise | 4137±805 | 5774±214 | 5137±225 | 5230±209 | **6747±427** |
| Medium Expert | Gravity | 4513±513 | 3748±892 | **4759±353** | 72±109 | **4707±779** |
| | Friction | 2684±2646 | 3748±892 | **9038±1480** | 7989±3999 | 6745±562 |
| | Joint Noise | 4137±805 | 3748±892 | **5288±104** | 733±767 | **5280±1329** |

both tasks, which serve as the offline dataset used in RL training. Additional real-world experiment settings like reward functions and dataset analyses are elaborated in Appendix B.2.

The control task is relatively challenging, as the robot has to use two wheels to keep balance and can easily fall to the ground. The simulation environment is constructed based on Isaac Gym [Makoviychuk *et al.*, 2021], depicted in Figure 2(b). The simulation environment suffers from intricate dynamics gaps. Notably, the electric motors in the real robot bear dead zones, which can lead to observational errors, whereas the simplified dynamics in simulation fails to capture this complicated situation. Besides, the rolling friction applied on wheels is well-simulated, while the sliding friction cannot be modeled realistically in Isaac Gym. Furthermore, the friction coefficient between wheels and the ground as well as its wear-and-tear, are not possible to be configured exactly as in the real-world environment. All these dynamics gaps can lead to serious sim-to-real transfer issues when deploying RL policy learned in simulation to real-world scenarios.

## 6.2 Baselines

As H2O is the first RL framework under the hybrid offline-and-online setting, we can only compare it with some purely online or offline RL algorithms, as well as representative methods that partially incorporate offline data in online or offline policy learning in a less integrated manner, which are:

- **SAC (sim)** [Haarnoja *et al.*, 2018]: the SOTA off-policy online RL algorithm, which is trained within the imperfect simulator and evaluated in the original simulation or real-world environments.

- **CQL (real)** [Kumar *et al.*, 2020]: the representative offline RL algorithm using value regularization. We run CQL on D4RL and recorded real-world robotic control dataset, which is not impacted by the dynamics gaps, but suffers from limited state-action space coverage in the offline datasets.

- **DARC** [Eysenbach *et al.*, 2020]: DARC uses a reward correction term derived from a pair of binary discriminators to optimize the policy within an imperfect simulator. We train these discriminators with real and simulation samples and learn policy within the simulation.

- **DARC+**: a variant of DARC, in which the real offline data are not only utilized for discriminator training, but also used in policy evaluation and improvement. DARC+ showcases the model behavior when we naïvely combine offline and online policy learning under the DARC framework.

## 6.3 Comparative Evaluation of H2O in Simulation and Real-World Experiments

In Table 1, we present the comparative results of H2O and other baselines on the simulation-based experiments. Notably, H2O achieves the best performance in almost all tasks compared with the baselines. Due to the relatively large dynamics gap, directly training SAC in simulation yields the poorest performance in most of the tasks. Offline RL method CQL achieves reasonable performance as compared with the online RL counterpart, which achieves second-highest scores in Medium and Medium Replay tasks. DARC performs surprisingly very well in the Medium Expert-Friction task, perhaps due to specific dataset and dynamics gap properties. DARC+ shows inferior performance compared with DARC in most tasks, indicating the insufficiency of naïvely combining offline and online policy learning. Nevertheless, H2O consistently provides superior performance under different offline datasets and dynamics gap settings, demonstrating the effectiveness of our proposed dynamics-aware policy evaluation scheme.

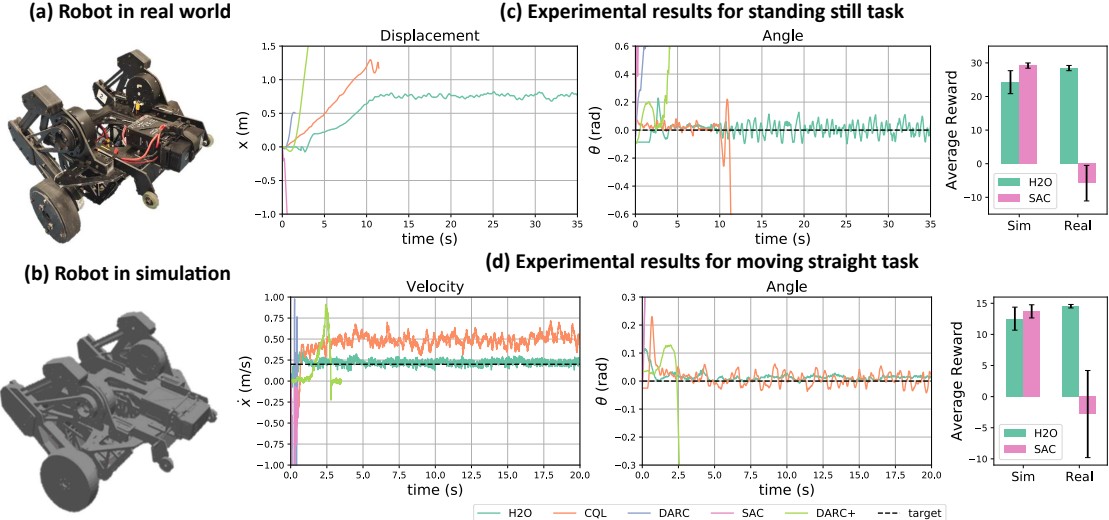

Figure 2: Real-world validation on a wheel-legged robot

In the real-world experiments[3], H2O manifests overwhelmingly better transferability on both standing still and moving straight tasks. In the standing still task, as demonstrated in the comparative curves in Figure 2(c), the robot with H2O policy remains steady after 11 seconds (s), while CQL bumps into the ground and goes out of control in 12s. DARC, DARC+, and SAC are not able to keep the robot balanced and quickly fail after initialization. In the moving straight task demonstrated in Figure 2(d), H2O achieves great control performance to keep balanced and closely follow the target velocity $v = 0.2$m/s, while CQL exceeds $v$ by a fairly large margin, nearly doubling the desired target velocity. Additionally, the angle $\theta$ illustrates that the robot controlled by H2O runs more smoothly than CQL. Once again, DARC, DARC+ and SAC fail at the beginning. Interestingly, we plot the evaluated average returns in both simulation and the real-world robot for SAC and H2O, revealing that achieving high performance in the simulation environment provides no guarantee for real-world transferability, if the simulator has a large sim-to-real gap. It is possible that policies with lower performance in the biased simulation could perform better in real-world scenarios. This suggests the potential need of reconsidering the widely adopted simulation-based evaluation or verification in mission-critical tasks, such as autonomous driving and industrial control, that overly trusting simulators with dynamics gaps can lead to serious consequences.

### 6.4 Ablation Study

In this section, we investigate the impact of each design component of H2O in the HalfCheetah-Medium Replay-Gravity task. In Table 2, we compare the performances of the variants of H2O by replacing adaptive value regularization weight $\omega$ into a uniform value, removing the importance weight in modified Bellman Error formulation $\widetilde{\mathcal{E}}$ or even removing the entire value regularization part from the framework. Without the adaptive value regularization, "H2O-a" demonstrates lower performance since the Q-values at high dynamics-gap samples are not properly regularized during training. Without the dynamics ratio as importance weight to fix the Bellman error, "H2O-dr", "H2O-a-dr" and "H2O-reg-dr" shows severe performance deterioration against their calibrated counterparts, which signifies the importance of correctly handling the Bellman updates when involving multiple sources of data. Removing the entire value regularization part from H2O leads to substantial performance degradation. However, it is noteworthy that "H2O-reg-dr" outperforms "H2O-dr", again suggests that fixing the Bellman error due to dynamics gap is very important in hybrid offline-and-online policy learning.

To further investigate the quality of the dynamics gap measure $u(\mathbf{s}, \mathbf{a})$ in H2O as well as its impact on the adaptive value regularization during the learning process, we construct a new test environment based on HalfCheetah. We apply specific noise offsets to actions based on X-velocity (see Figure 3) in the simulation, such that larger X-velocity bears larger noise offsets. We plot the dynamics gap measure $u(\mathbf{s}, \mathbf{a})$ and the corresponding Q-values $Q(\mathbf{s}, \mathbf{a})$ for each simulated sample $(\mathbf{s}, \mathbf{a})$ from a

---

[3]Please refer to supplementary video for performances of H2O and baselines on the real wheel-legged robot.

Table 2: Ablations on H2O. "-a" denotes no adaptive weights $\omega$. "-dr" denotes no dynamics ratio to fix the Bellman error. "-reg" stands for no value regularization applied in the framework.

| | H2O | H2O-a | H2O-dr | H2O-a-dr | H2O-reg | H2O-reg-dr | CQL | SAC |
|---|---|---|---|---|---|---|---|---|
| Adaptive $\omega$ | ✓ | ✗ | ✓ | ✗ | - | - | - | - |
| Uniform $\omega$ | ✗ | ✓ | ✗ | ✓ | - | - | - | - |
| Modified $\widetilde{\mathcal{E}}\left(Q, \hat{\mathcal{B}}^{\pi}\hat{Q}\right)$ | ✓ | ✓ | ✗ | ✗ | ✓ | ✗ | - | - |
| Average Return | **6813±289** | 6675±179 | 4721±196 | 5223±198 | 6501±147 | 5290±356 | 5774±214 | 4513±513 |

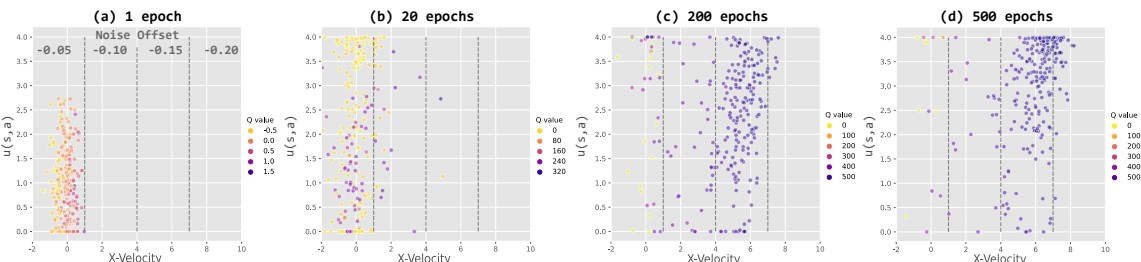

Figure 3: The dynamics gap measure $u(\mathbf{s}, \mathbf{a})$ evaluated during the training process

batch in some stages of the learning process. Notably, in the HalfCheetah task, higher X-velocity corresponds to higher rewards and hence corresponds to higher Q-values, which at the same time may suffer from more severe dynamics gaps. We observe that as the course of learning proceeds, the dynamics gap measures $u$ in H2O indeed capture the higher dynamics gap information at the later part of the training, and works as expected to penalize the Q-values on high dynamics-gap samples.

# 7 Conclusion and Perspectives

In this paper, we propose the dynamics-aware Hybrid Offline-and-Online RL (H2O) framework to combine offline and online RL policy learning, while simultaneously addressing the sim-to-real dynamics gaps in an imperfect simulator. H2O introduces a dynamics-aware policy evaluation scheme, which adaptively penalizes the Q-values as well as fixes the Bellman error on simulated samples with large dynamics gaps. This scheme can be shown as equivalent to a special kind of reward adjustment, which places reward penalties on high dynamics-gap samples, but boosts reward in regions with low dynamics gaps and abundant offline data. Under H2O, both the offline dataset and the imperfect simulator are fully used and complement each other to extract the maximum information for policy learning. In particular, the limited size and state-action space coverage of the offline datasets are greatly supplemented by unrestricted simulated samples; the dynamics gaps in simulated samples can also be alleviated during policy learning given correct dynamics information from the real data. Through extensive simulation and real-world experiments, we demonstrate the superior performance of H2O against other cross-domain online and offline RL methods.

Online and offline RL have been two separately studied fields in the past, and both face a set of specific challenges. Few attempts have been made to organically combine these two different RL paradigms. H2O provides a brand new hybrid offline-and-online RL paradigm, which shows promising results when leveraging offline data and simulation-based online learning in an integrated framework. Still, there are several future works that can be done to further enhance the model performance, such as adopting a less conservative offline RL backbone algorithm as compared to CQL, and incorporating better dynamics gap quantification methods that do not need approximation. We hope the insights developed in this work can shed light on future hybrid RL algorithm design, which can potentially provide better RL solutions for practical deployment.

## Acknowledgments and Disclosure of Funding

This work is supported by funding from Haomo.AI. Haoyi Niu is also funded by Tsinghua Undergraduate "Future Scholar" Scientific Research Grant, i.e. Tsinghua University Initiative Scientific Research Program (20217020007). The authors would also like to thank the anonymous reviewers for their feedback on the manuscripts.

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
