# OpenReview forum: "When to Trust Your Simulator: Dynamics-Aware Hybrid Offline-and-Online Reinforcement Learning"
_NeurIPS.cc/2022/Conference — NeurIPS 2022 Accept_

### Official Review · Reviewer_X8Vb · 2022-07-06

**Rating:** 4
**Confidence:** 4
**Soundness:** 2 fair
**Presentation:** 2 fair
**Contribution:** 2 fair

**Summary:**

This paper proposes a novel algorithm of offline-to-online reinforcement learning for *domain-shift* between offline datasets and online executable environments. This might be an important setting for addressing real-world tasks, since we could typically access a limited amount of real-world data and easily accessible simulators. Since the source online executable simulators and the target offline data have a large dynamics gap, the proposed method, H2O, employs (1) adaptive value regularization for high dynamics-gap transitions and (2) dynamics-corrected Bellman error. These regularizations on the objective functions correspond to penalty rewards that punish highly dynamics-gapped samples in a tablular case. The experiments show transfers from the offline dataset on a normal simulator to unreal dynamics simulator in MuJoCo HalfCheetah, and a sim-to-real transfer with the wheel-legged robot.

**Questions:**

- Checklist 2 (a) is required to be fixed.
- $\beta$ in Eq 4 seems duplicated to a parameter of behavioral policy $\pi_{\beta}$.
- TD3-BC (Fujimoto et al. [5]) might be a competitive baseline (BC with offline data, and optimizing TD3 term with simulator transitions).
- Also, an offline model-based method such as MOPO may deal with this setting, by training the dynamics model with real-world offline data and training the policy within imaginary rollouts with a pessimistic penalty.
- Citation of Liu et al. [2] seems to be presented in ICLR 2022, not ICLR2021.

---
[4] Fujimoto et al. A Minimalist Approach to Offline Reinforcement Learning. 2021.


**Limitations:**

- The proposed method seems to highly depend on CQL, compared to DARA [2].
- The empirical evaluation is only done on HalfCheetah, which seems very limited, comapred to prior works [1, 2].


**Strengths And Weaknesses:**

[S] ... Strengths, [W] ... Weaknesses

### Originality
- [S] Offline-to-online reinforcement learning for domain-shift has not been explored yet, but seems an important problem for real-world applications.
- [W] H2O seems highly co-dependent to CQL and DARC (Eysenbach et al. [1]), while relevant work in offline RL, DARA (Liu et al. [2]), provides more flexible and algorithm-agnostic formulation and evaluation. The difference between CQL and H2O is dynamics-gap-based reweighting $\omega(s, a)$ compared to CQL(\mathcal{H}), which uses uniform action distribution for the weighted sum of $\exp(Q)$.

### Quality
- [S] This paper is well-structured and easy to follow.
- [W] The experimental coverage might be limited because H2O is only tested on HalfCheetah (Table 1).
- [W] I'm a bit confused about experimental settings on the simulator, described in Section 6.1. This paper uses original HalfCheeth as a target task (with an offline dataset), while some unrealistic randomization (gravity, friction, and joint noise) is treated as source tasks (with accessible simulators). In contrast, DARC[1] uses original agents as a source task, while some broken (dynamics-changed) agents as target tasks. I'm not sure why the authors flip the settings, and whether H2O works in a reverse setting. In my opinion, even if we could not model the real-world dynamics perfectly, we might use the best-efforted simulator for training. So the setting in DARC[1] seems more likely than that of this paper in the real-world problem.

### Clarity
- [W] I'm not sure whether density ratio estimation (via discriminators) in Eq 7 is needed or not, since H2O directly models transition dynamics from "real" offline datasets (Section 4.4) $P_{\mathcal{\hat{M}}}$, with Gaussian distribution as typical model-based RL method did. So it seems a straightforward but effective approach to model simulator "source" dynamics $P_{\mathcal{M}}$ with Gaussian distribution.
- [W] As for Figure 1 (Average Reward Bar graph), it might be valuable to include the results of other baselines, such as CQL and DARC(+). As far as I checked Appendix, there are no additional details.

### Significance
- [S] This work shows good sim-to-real finetuning results between IssacGym and real wheel-legged robots, which might be valuable to prove the proposed method might be directly connected to real-world application.
- [W] I'm not fully sure that this work would be properly placed following previous sim-to-real literature. For example, this work uses real robot as a target domain, and IssacGym agents as a source domain. Since IsaacGym has a well-designed dynamics-parameter-randomization [3], and such simple parameter-randomized training helps the agent to successfully deal with real-world dynamics [4]. It might be beneficial to include such dynamics randomization as a baseline.
- [W] While the authors say, "domain randomization often needs manually-specified randomized parameters and nuanced randomization distributions (L77, 78)", the choice of $\omega(s,a)$ in H2O is also a heuristic. So randomized dynamics training might be a fair comparison against H2O.

---
[1] Eysenbach et al. Off-Dynamics Reinforcement Learning: Training for Transfer with Domain Classifiers. 2021.

[2] Liu et al. DARA: Dynamics-Aware Reward Augmentation in Offline Reinforcement Learning. 2022.

[3] Makoviychuk et al. Isaac Gym: High Performance GPU Based Physics Simulation For Robot Learning. 2021.

[4] Chen et al. A System for General In-Hand Object Re-Orientation. 2021.

---

> ### Author Response · Authors · 2022-08-02
> **Response to Reviewer X8Vb (5/5)**
>
> ## Response to Comments in Significance
>
> > **(8) "It might be beneficial to include such dynamics randomization as a baseline....the choice of ω(s,a) in H2O is also a heuristic. So randomized dynamics training might be a fair comparison against H2O"**
>
> **RESPONSE:**
>
> * We have to point out that $\omega$ is estimated by KL divergence of simulation and real dynamics based on the results of domain discriminators, which is considerably less heuristic as compared to domain randomization. Domain randomization techniques blindly randomize the simulation parameters and often need manually-specified randomized parameters and nuanced randomization distributions. By contrast, H2O addresses the dynamics gap in a more specific manner as it directly models the dynamics gap between the real and simulation environments. Moreover, the only hyperparameter used in H2O for evaluating the dynamics gap distribution is the number of resampled next states ($N$, see Eq. 28). In all our experiments, we fixed it to 10 without any hyperparameter tuning and this choice has led to reasonably good performance for all tasks.
> * Besides, the philosophy of H2O is addressing sim2real issues in an integrated RL learning process, rather than adding additional steps (e.g. domain randomization) in RL training or relying on other treatments.
>
> ## Response to Questions
>
> > **(9) " Questions: Checklist 2 (a) is required to be fixed. β in Eq 4 seems duplicated to a parameter of behavioral policy πβ."**
>
> **RESPONSE:** We thank the reviewer for the comment. We have revised the notation of behavior policy as $\pi_{\mathcal{D}}$ in the revised version of the paper.
>
>
> > **(10) " Questions: TD3-BC (Fujimoto et al. [5]) might be a competitive baseline ... MOPO may deal with this setting..."**
>
> **RESPONSE:** We thank the reviewer for the comment.
>
> * Our experiments mainly focused on comparing with cross-domain RL methods rather than comparing with pure offline RL algorithms. We include CQL as a baseline because it is the backbone algorithm of H2O and thus can be used for ablation.
> * The performance of MOPO is shown to be inferior to COMBO [6], which resembled one of the variants of H2O (H2O-a-dr) in our ablation study. In this variant, we use the simulator as the model in COMBO for data generation, but remove the adaptive weights and correction of Bellman error. Based on our ablation results in Table 2, we find that H2O-a-dr actually performs a lot worse than H2O. The reason is that model-based offline RL methods are not specifically designed for handling dynamics gaps between different domains, thus suffer from performance drop.

---

> ### Author Response · Authors · 2022-08-03
> **Response to Reviewer X8Vb (4/5)**
>
> * First, the transition dynamics of "real" offline datasets is $P_\mathcal{M}$ in our paper rather than $P_\mathcal{\widehat{M}}$ mentioned by the reviewer, the latter is the dynamics of the simulator.
> * Second, we avoid making assumptions about the simulator and treat it as a black-box model. We follow the most generic setting that the agent interacts with the simulator by only observing the current state $s$ and provide an action $a$.
>   In this case, it is typically not feasible to evaluate $s'\sim P_{\widehat{\mathcal{M}}}\left(\mathbf{s}^{\prime} \mid \mathbf{\bar{s}}, \mathbf{a}\right)$ given arbitrary state $\bar{s}$, since it will require the simulator to have the capability to reset the simulation to a given state $s$. Thus in order to evaluate the
>
> $s'\sim P_{\widehat{\mathcal{M}}}\left(\mathbf{s}^{\prime} \mid \mathbf{\bar{s}}, \mathbf{a}\right)$ in the sampling and summation part of the dynamics gap $u(\mathbf{s}, \mathbf{a}):=D_{K L}\left(P_{\widehat{\mathcal{M}}} \| P_{\mathcal{M}}\right) \approx \sum_{\mathbf{s}^{\prime} \sim P_{\widehat{\mathcal{M}}}\left(\mathbf{s}^{\prime} \mid \mathbf{s}, \mathbf{a}\right)}^{N} \log \frac{P_{\widehat{\mathcal{M}}}\left(\mathbf{s}^{\prime} \mid \mathbf{s}, \mathbf{a}\right)}{P_{\mathcal{M}}\left(\mathbf{s}^{\prime} \mid \mathbf{s}, \mathbf{a}\right)}$,
>
> we adopt the cheapest method to approximate
>
> $P_{\widehat{\mathcal{M}}}\left(\mathbf{s}_{i}^{\prime} \mid \mathbf{s}, \mathbf{a}\right)=$
> $\mathcal{N}\left(\mathbf{s}^{\prime}, \hat{\Sigma}_{\mathcal{D}}\right)$
>
> for sampling $s^{\prime}$ given $(s, a, s^{\prime})$ from the simulated replay buffer $B$ (see description in Section 4.4). We find this treatment although simple, but sufficient for H2O to achieve good performance. However, if we use the same approximation for $P_{\widehat{\mathcal{M}}}$ to compute the dynamics ratio
>
> $\frac{P_{\widehat{\mathcal{M}}}\left(\mathbf{s}^{\prime} \mid \mathbf{s}, \mathbf{a}\right)}{P_{\mathcal{M}}\left(\mathbf{s}^{\prime} \mid \mathbf{s}, \mathbf{a}\right)}$,
>
> It will cause an overly large approximation error and damage the model performance. As a result, we choose to use two discriminators to approximate the dynamics ratio in a sample-based manner, rather than directly model and evaluate the ratio of two distributions.
> * Lastly, we indeed tested another variant of H2O which uses a learned dynamics model
>
> $P_{\widetilde{\mathcal{M}}}\left(\mathbf{s}_{i}^{\prime} \mid \mathbf{s}, \mathbf{a}\right)$
>
> from offline data as in model-based offline RL methods as mentioned by the reviewer, and then use the reverse-KL to estimate the dynamics gap (see Eq. 31 in Appendix C.3). In this implementation, we use the deep neural network to learn a probabilistic model
>
> $P_{\widetilde{\mathcal{M}}}$ that approximates $P_{\mathcal{M}}$ similar to MOPO and COMBO. The next state $s^{\prime}$ is directly sampled from $P_{\widetilde{\mathcal{M}}}$, and we again approximate the dynamics ratio using the same two discriminators. The final performance of this variant does not show noticeable performance improvement (see the table below and Appendix C.1 in the update paper), at the extra cost of learning an additional dynamics model. As a result, we did not include this variant in our paper.
>
>
> | HalfCheeetah_Gravity                   |  Medium  | Medium-Replay | Medium-Expert |
> | :------------------------------------- | :------: | :-----------: | :-----------: |
> | H2O-KL (original version in the paper) | 7085±416 |   6813±289    |   4707±779    |
> | H2O-Reverse KL                         | 7065±170 |   6476±129    |   4709±274    |
>
> > **(7) "As for Figure 1 (Average Reward Bar graph), it might be valuable to include the results of other baselines, such as CQL and DARC(+)."**
>
> **RESPONSE:** We thank the reviewer for the comment. Originally, we intend to use the bar plots of SAC and H2O in both domains to reveal that achieving high performance in the simulation environment provides no guarantee for real-world transferability. We have included the figure for cumulative rewards of DARC, DARC+, and CQL from the logged data in the real-world experiments. Please check it in Appendix B.3 and Figure 4 in the revised version of the paper.

---

> ### Author Response · Authors · 2022-08-03
> **Response to Reviewer X8Vb (3/5)**
>
> ||DARC [6]|DARA [7]|H2O|
> |:--------:|:--------:|:--------:|:--------:|
> |RL paradigm|Online|Offline|Offline-and-Online|
> |Validation|simulation|simulation + real-world|simulation + real-world|
> |Online simulation interaction|√|×|√|
> |Online real system interaction|√|×|×|
> |Real-world data for policy learning|×|√|√|
>
> [5] Aviral Kumar et al. Conservative Q-Learning for Offline Reinforcement Learning. NeurIPS 2020.
>
> [6] Tianhe Yu et al. COMBO: Conservative Offline Model-Based Policy Optimization. NeurIPS 2021.
>
> [7] Jinning Li et al. Dealing with the Unknown: Pessimistic Offline Reinforcement Learning. CoRL 2021.
>
> [8] Haoran Xu et al. Constraints penalized q-learning for safe offline reinforcement learning. AAAI 2022.
>
> [9] Eysenbach et al. Off-Dynamics Reinforcement Learning: Training for Transfer with Domain Classifiers. ICLR 2021.
>
> [10] Liu et al. DARA: Dynamics-Aware Reward Augmentation in Offline Reinforcement Learning. ICLR 2022.
>
> ## Response to Comments in Quality
> > **(4) "The experimental coverage might be limited because H2O is only tested on HalfCheetah (Table 1)"**
>
> **RESPONSE:** This statement is not accurate since H2O is also tested on a real-world wheel-legged robot. We have conducted additional experiments on Walker2d with Medium Replay dataset during the limited time of rebuttal. Results are presented below, with exactly the same hyperparameter setting as that of HalfCheetah tasks in our paper. It is found the H2O again achieves the best performance among all other baselines. We have included these experimental results in Appendix C.2 of the revised paper.
>
> | Unreal Dynamics|SAC|CQL|DARC|DARC+|H2O|
> |:-----:|:-----:|:-----:|:-----:|:-----:|:-----:|
> |Gravity|1233±841|1445±1077|1987±965|1618±1446|2187±1103|
> | Friction|2879±569|1445±1077|2518±1244|2375±579|3656±582|
> |Joint Noise|852±386|1445±1077|64±115|630±561|2998±854|
>
> > **(5) "I'm a bit confused about experimental settings on the simulator, described in Section 6.1. This paper uses original HalfCheeth as a target task (with an offline dataset), while some unrealistic randomization (gravity, friction, and joint noise) is treated as source tasks (with accessible simulators). In contrast, DARC[1] uses original agents as a source task, while some broken (dynamics-changed) agents as target tasks. I'm not sure why the authors flip the settings, and whether H2O works in a reverse setting. In my opinion, even if we could not model the real-world dynamics perfectly, we might use the best-efforted simulator for training. So the setting in DARC[1] seems more likely than that of this paper in the real-world problem."**
>
> **RESPONSE:**
> * The primary reason that we choose the original MuJoCo environment as the target domain is to use the standard benchmark datasets in D4RL (Fu et al., 2020) for model evaluation. Using the D4RL datasets as real offline data, the evaluated model performance is directly comparable to the vast literature in the area of offline RL. However, if we handcraft a "broken agent" as the target task, then it will make the readers hard to compare our model performance with pure offline RL approaches.
> * Second, in most real-world application scenarios, real systems are far more complex than abstracted simulation environments. Due to complex internal dynamics or unobserved information, building a high-fidelity simulator can be extremely costly, and even existing the best-effort simulator may still suffer from severe sim2real transfer issues. Critically, our method achieves overwhelming performance against DARC and other baselines on the real-world wheel-legged robot, which demonstrates the applicability of H2O to real-world problems. In other words, simulation environments empirically bear flawed dynamics upon the original (real-world) settings.
> ## Response to Comments in Clarity
> > **(6) "I'm not sure whether density ratio estimation (via discriminators) in Eq 7 is needed or not, since H2O directly models transition dynamics from "real" offline datasets (Section 4.4) PM^, with Gaussian distribution as typical model-based RL method did. So it seems a straightforward but effective approach to model simulator "source" dynamics PM with Gaussian distribution."**
>
> **RESPONSE:** The dynamics ratio estimation is necessary for H2O. (see the following)

---

> ### Author Response · Authors · 2022-08-03
> **Response to Reviewer X8Vb (2/5)**
>
> ## Regarding the Originality
>
> > **(3) "H2O seems highly co-dependent to CQL and DARC (Eysenbach et al. [1]), while relevant work in offline RL, DARA (Liu et al. [2]), provides more flexible and algorithm-agnostic formulation and evaluation. The difference between CQL and H2O is dynamics-gap-based reweighting ω(s,a) compared to CQL(\mathcal{H}), which uses uniform action distribution for the weighted sum of exp⁡(Q)"**
>
> **RESPONSE:**
>
> * **Regarding the relationship to CQL:** CQL provides a general value regularization framework and has been adopted in a rich amount of methods, such as COMBO [6], PessORL [7], and CPQ [8]. Most of these algorithms are designed for offline settings and focus on regularizing OOD actions or states. Although H2O also adopts the CQL framework, it has several key differences from CQL and its existing extensions. H2O is the first algorithm that provides a hybrid offline-and-online RL paradigm to tackle the sim2real issue when jointly learning from a biased simulator and offline real data. By this design, H2O uses a very different value penalization scheme, which focuses on penalizing the dynamics gap rather than OOD states or actions. Moreover, unlike all other CQL-style algorithms, H2O does not aim at performing conservatism but adaptation (see theoretical analysis in Section 5 and Appendix A.2). We summarize the key distinctions of H2O against other CQL-style algorithms in the following table.
>
> | |CQL [5]|        COMBO [6]|PessORL[7]|          CPQ[8] | H2O|
> |:--------:|:--------:|:--------:|:--------:|:--------:|:--------:|
> |RL paradigm|Offline|Offline|Offline|Offline constrained |Offline-and-online|
> |Dynamics gap| None | Partial (due to model approximation error) | None | None | Yes
> |Regularization penalty|OOD actions|OOD actions|OOD states|OOD actions and constraints violations|state-action pairs with high dynamics gap|
> |Validation|simulation|simulation|simulation|simulation|simulation + real-world|
>
> * **Regarding the relationship with DARC:**
>   * The only similar component between H2O and DARC/DARA is the use of trained discriminators to compute the dynamics ratio. Despite this, H2O and DARC/DARA are developed from very different angles. The theoretical foundation of DARC is built upon minimizing the divergence of trajectory distributions between source and target domains, and then deriving the reward corrective term $\Delta r$ to handle the dynamics gap. DARA is the offline version of DARC which shares lots of similarities of DARC in its practical algorithm (the forms of $\Delta r$ in DARC and DARA are identical). The downside of this theoretical foundation is that it implicitly assumes the offline data to have relatively high quality, otherwise minimizing the trajectory divergence between low-performance real data and the policy-generated distribution will lead to suboptimal policies. We have empirically observed (see Table 1 in the paper) that DARC-style methods struggle in tasks with low-quality real-world data (Medium and Medium Replay datasets). By contrast, H2O is developed under a completely different value regularization framework, which does not suffer from this problem.
>   * Moreover, DARC leverages the real-world samples only to train discriminators, giving up the valuable information in real samples during policy evaluation. DARA although assumes access to the simulator, but disappointedly sacrifices the unrestricted online exploration in the simulated domain, which could lead to low sample efficiency when utilizing the simulated data. None of them is able to develop an integrated framework like H2O by combining offline data with online learning and simultaneously addressing the dynamics gap. We also summarize the differences between H2O and DARC/DARA in the following table.

---

> ### Author Response · Authors · 2022-08-03
> **Response to Reviewer X8Vb (1/5)**
>
> We thank the reviewer for the thorough and detailed comments. Regarding the concerns from the reviewer, we describe the detailed response as follows:
>
> ## Regarding the Summary of the Reviewer X8Vb
>
> > **(1) "This paper proposes a novel algorithm of offline-to-online reinforcement learning for domain-shift between offline datasets and online executable environments."**
>
> **RESPONSE:** As we have indicated in the title, abstract, and introduction of this paper, the proposed hybrid offline-and-online RL algorithm H2O, as well as its problem setting, are distinguished from the "offline-to-online RL" [1-4] mentioned by the reviewer in two aspects:
>
> * **RL paradigm**: Offline-to-online RL methods in literature [1-4] adopt a two-stage learning process, i.e., first run an offline RL/IL algorithm to obtain a good initial policy, and then run an online off-policy RL algorithm for online fine-tuning. Two sub-algorithms are involved in offline-to-online RL. By contrast, H2O introduces a brand-new hybrid RL paradigm, which simultaneously learns from offline and online data from two domains in every training step in a single integrated learning process.
>
> * **Application scenario**: Offline-to-online RL methods consider a single real environment, but with offline and online data. Hence there is no dynamics gap or the need to consider the sim2real issue. However, H2O considers a different sim2real setting, where we have a biased simulator and the real-world system, and the data are from two domains with dynamics gaps. Furthermore, unlike the typical offline-to-online setting that allows real system interaction during the online fine-tuning stage, H2O does not allow real system interaction during training.
>
> We summarize the key differences between offline-to-online and H2O in the following table:
>
> |                         |            Learning process            |  Environments   | Allow real system interaction |        Handling Distribution Shift         | Handling Dynamics Gap |
> | :---------------------: | :------------------------------------: | :--------------: | :---------------------------: | :----------------------------------------: | :-------------------: |
> |           H2O           |                 Single                 | 2 (sim and real) |              No               |         √ (caused by domain shift)         |           √           |
> | Offline-to-online [1-4] | Two-stage (first offline, then online) |  1 (only real)   |              Yes              | √ (caused by offline-to-online transition) |           ×           |
>
>
> [1] Seunghyun Lee et al. Offline-to-Online Reinforcement Learning via Balanced Replay and Pessimistic Q-Ensemble. CoRL 2021.
>
> [2] Tengyang Xie et al. Policy Finetuning: Bridging Sample-Efficient Offline and Online Reinforcement Learning. NeurIPS 2021.
>
> [3] Yi Zhao et al. Adaptive Behavior Cloning Regularization for Stable Offline-to-Online Reinforcement Learning. Offline RL Workshop at NeurIPS 2021.
>
> [4] Desik Rengarajan et al. Reinforcement Learning with Sparse Rewards using Guidance from Offline Demonstration. ICLR 2022.
>
> > **(2) "These regularizations on the objective functions correspond to penalty rewards that punish highly dynamics-gapped samples in a tabular case."**
>
> **RESPONSE:** First, the adaptive value regularization of H2O not only penalizes reward for high dynamics-gapped samples but could also boost reward for simulation data with low dynamics gap or have sufficient information from offline data (see discussion in Section 5). Second, although the theoretical analysis of H2O is derived in a tabular case, its form and implementation are also applicable to continuous state-action space. If we replace the summation over $(s,a)$ to integration in Eq.(5)-(6), the formulations of H2O can be ported to the continuous state-action space settings. In fact, our experiments on MuJoCo-HalfCheetah and wheel-legged robot have all confirmed the good performance of H2O on continuous control tasks.

---

> ### Comment · Reviewer_X8Vb · 2022-08-06
> **Response to the rebuttal (1/2)**
>
> I appreciate the authors answering my questions and addressing several concerns I raised, as well as additional experiments. Further feedbacks and flags are below. I hope these will help to improve your work.
>
> **Re: (1)**
>
> Now, the difference between offline-to-online RL and offline-and-online RL is clear to me, they seem to have similar but distinct problem formulations. The one thing I came up with after reading the response is whether "Hybrid Offline and Online" describes what this paper address well. For example, [1,2,3] can be called a "Hybrid Offline and Online" RL even though they tackle single dynamics since they leverage offline demonstration to accelerate online training. The key of this paper seems to be "dynamics-awareness", "off-dynamics", or "sim-to-real", or else. So it might help the comprehensibility that those words are included in the proposed term.
>
> **Re: (2)**
>
> Thanks for the clarification. What I wanted to point out here was, L228 said "Assuming that the Q function is tabular, we can find the Q function corresponding to the new objective ...". If these analyses are easily applicable to continuous control, these statements should be included in the main text explicitly (or Appendix). Unfortunately, I could not find some notes or links to the appendix. I guess the tabular case analysis also works well in a practical setting in some cases, but the rigorous analysis may have some obstacles.
>
> **Re: (3)**
>
> I appreciated the authors for clarification about the relationship among previous offline works (CQL, COMBO, PessORL, and CPQ) and off-dynamics works (DARC, DARA). Since H2O also could be considered as having the reward corrective term  to handle the dynamics gap (Eq. 11), I think it would be helpful to include short note about how they are different in the related work section.
>
> +1
> > We have empirically observed (see Table 1 in the paper) that DARC-style methods struggle in tasks with low-quality real-world data (Medium and Medium Replay datasets). By contrast, H2O is developed under a completely different value regularization framework, which does not suffer from this problem.
>
> I think this should be carefully stated, since the collected-demonstration quality in the real-world experiment seems high (Fig 3). Whether H2O does not suffer from low-quality real-world data does not seem to be evaluated.
>
> **Re: (4)**
>
> To be precise, it should be "The coverage in  Simulation-based experiment might be limited because H2O is only tested on HalfCheetah (Table 1).". I apologize for inaccurate sentence. I checked the results of Walker2d and it seem to support the main claim.
>
> **Re: (5)**
>
> In my understanding and according to the author's response (1), this paper doesn't intend to compare the performance against pure offline RL, since this paper studies offline-and-online RL on two different dynamics (since offline RL can't access any simulator, the comparison might not be fair). Also, the performances were not normalized, while almost all recent offline RL works are, and the environments the authors used were limited to halfcheetah + a part of walker2d.
>
> > Our experiments mainly focused on comparing with cross-domain RL methods rather than comparing with pure offline RL algorithms. We include CQL as a baseline because it is the backbone algorithm of H2O and thus can be used for ablation.
>
> Moreover, the design of source tasks doesn't seem to be realistic. In gravity tasks, source "sim" environments have $2g$ gravity acceleration. I don't think no one uses such an unrealistic simulator for training. In joint noise tasks, adding gaussian noise N(0, 1) seems too large, since the action space of MuJoCo is [-1, 1]. Since this range is equivalent to the scale parameter, each action dimension can be easily crushed. I think such situations or controllers rarely happen even in the real world. Friction tasks seem relatively decent, since modeling friction accurately might be a difficult task. However, I'm not sure if x0.3 friction is proper, and no other values are tested (probably x1.1 or x0.9 or at random at each step are more realistic). Since the perturbation parameters in source "sim" environments may affect the algorithm performances largely, a more detailed analysis might improve this paper that aims for real-world applications.
>
> I agree that the experiments with a real-world wheel-legged robot are a great result, but the literature review of sim-to-real works is limited (in section 2.1), and no baselines in the experiment (I think DARC and DARA are not sim-to-real work). Actually, some works [2,3] propose demonstration-guided sim2real algorithms. I think it is of course difficult to include them directly as baselines, but as I mentioned in the review, a simple randomization strategy might be a good baseline.
>
> **Re: (6)**
>
> I also appreciated the authors for clarification about the dynamics ratio estimation, and additional experiments. These seem to show H2O chose a decent strategy.

---

> > ### Comment · Reviewer_X8Vb · 2022-08-06
> > **Response to the rebuttal (2/2)**
> >
> > **Re: (7)**
> >
> > I checked those figures in the revised paper. This would be helpful. Maybe the scale of reward between Appendix B.3 and Figure 4 should be aligned.
> >
> > **Re: (8)**
> >
> > Thanks for your clarification. The choice of $\omega$ becomes clear to me now.
> >
> > **Re: (10)**
> >
> > I appreciate the authors' detailed comments. The reason about MOPO is convincing. The reason why I raise TD3+BC as a potential baseline is a significant relevance to demonstration-guided RL. Combining demonstration to RL is well-studied and has rich previous literature (e.g. [1, 4]) even in sim2real domain [2, 3].
> >
> > ---
> > [1] Rajeswaran et al. Learning Complex Dexterous Manipulation with Deep Reinforcement Learning and Demonstrations. 2017.
> >
> > [2] Zhu et al. Reinforcement and Imitation Learning for Diverse Visuomotor Skills. 2018.
> >
> > [3] Matas et al. Sim-to-Real Reinforcement Learning for Deformable Object Manipulation. 2018.
> >
> > [4] Hester et al. Deep Q-learning from Demonstrations. 2017.

---

> > > ### Author Response · Authors · 2022-08-08
> > > **Response to the follow-up questions (3/3)**
> > >
> > > ## Response to "Re:(10)"
> > > > **"The reason why I raise TD3+BC as a potential baseline is a significant relevance to demonstration-guided RL."**
> > >
> > > **RESPONSE:** The hybrid TD3+BC baseline proposed by the reviewer ("BC with offline data, and optimizing TD3 term with simulator transitions") could suffer from severe dynamics shift problems and can easily mislead the learning process. Note that the policy maximization procedure under the hybrid TD3+BC baseline mentioned by the reviewer has the following form:
> > >
> > > $$\pi = \arg\max_{\pi} \left[E_{(s,a)\sim B} \lambda Q(s,\pi(s)) - E_{(\hat{s},\hat{a})\sim D}(\pi(\hat{s})-\hat{a})^2\right]$$
> > >
> > > where $B$ is the simulation buffer and $D$ is the real-world data. In this case, the state-action pairs used for max-Q and the BC regularization are from different datasets $\left((s,a)\sim B \text{ vs } (\hat{s},\hat{a})\sim D\right)$. In this case, the BC regularizer does not regularize $\pi(s)$ on state $s$ explored in online learning, but on some unrelated state $\hat{s}$ and the corresponding actions $\hat{a}$. This makes maximizing Q and minimizing BC loss becomes two completely independent and conflicting objectives, and can lead to severe dynamics shift issues on learned policies.
> > >
> > > The original offline TD3+BC will not have this issue, as all $(s, a, s^\prime)$ triples are from the offline dataset $D$, and the BC loss regularizes directly between $\pi(s)$ and the corresponding action $a$ in data:
> > >
> > > $$\pi = \arg\max \left[E_{(s,a)\sim D} \lambda Q(s,\pi(s)) - (\pi(s)-a)^2\right]$$
> > >
> > > If the reviewer insists on this potential baseline, we will consider adding the results in the final version of our paper.

---

> > > > ### Author Response · Authors · 2022-08-09
> > > > **Follow-up Response to Reviewer X8Vb**
> > > >
> > > > We have completed another set of experiments on HalfCheetah-Gravity task with D4RL random datasets, as well as additional comparative experiments with the hybrid TD3+BC baseline mentioned by the reviewer. We summarize the detailed results as follows.
> > > >
> > > > ## Additional experiments to demonstrate the performance of H2O over DARC-style methods (*Response to "Re(3)"*)
> > > >
> > > > **RESPONSE:** We have conducted the experiments on HalfCheetah-Gravity tasks with D4RL random dataset in addition to our previous results on HalfCheetah-Friction-Random task. The results are consistent with our previous conclusion that H2O does not suffer from low-quality real-world data but DARC-style methods do.
> > > >
> > > > |Dataset| Unreal Dynamics|SAC|CQL|DARC|DARC+|H2O|
> > > > |:-----:|:-----:|:-----:|:-----:|:-----:|:-----:|:-----:|
> > > > |HalfCheetah_Random|Gravity |4513±513|2465±180|357±617|-97±121|**4602±223**|
> > > > |HalfCheetah_Random|Friction|2684±2646|2465±180|537±250|425±99|**4862±1608**|
> > > >
> > > > ## Additional Results for hybrid TD3+BC baseline (*Response to "Re(10)"*)
> > > >
> > > > **RESPONSE:** We have discussed the potential infeasibility of the hybrid TD3+BC ("BC with offline data, and optimizing TD3 term with simulator transitions") as a competitive baseline in our previous response. Here we report the detailed performance scores of hybrid TD3+BC baseline (averaged over 3 seeds) and H2O in HalfCheetah environment with different quality of D4RL datasets and different types of unreal dynamics properties. The results are presented in the table below, revealing the extremely poor performance of hybrid TD3+BC.
> > > >
> > > > |Dataset|Unreal Dynamics|TD3+BC|H2O|
> > > > |:-----:|:-----:|:-----:|:-----:|
> > > > |HalfCheetah_Medium|Gravity|-407±81|**7085±416**|
> > > > |HalfCheetah_Medium|Friction|-482±102|**6848±445**|
> > > > |HalfCheetah_Medium|Joint Noise|-396±105|**7212±236**|
> > > > |HalfCheetah_Medium-Replay|Gravity|73±576|**6813±289**|
> > > > |HalfCheetah_Medium-Replay|Friction|-493±58|**5928±896**|
> > > > |HalfCheetah_Medium-Replay|Joint Noise|100±834|**6747±427**|
> > > > |HalfCheetah_Medium-Expert|Gravity|-385±150|**4707±779**|
> > > > |HalfCheetah_Mediume-Expert|Friction|-504±24|**6745±562**|
> > > > |HalfCheetah_Medium-Expert|Joint Noise|-478±21|**5280±1329**|

---

> > > ### Author Response · Authors · 2022-08-08
> > > **Response to the follow-up questions (2/3)**
> > >
> > > ## Response to "Re:(5)"
> > > > **"In my understanding and according to the author's response (1), this paper doesn't intend to compare the performance against pure offline RL, since this paper studies offline-and-online RL on two different dynamics (since offline RL can't access any simulator, the comparison might not be fair). Also, the performances were not normalized, while almost all recent offline RL works are..."**
> > > > **Moreover, the design of source tasks doesn't seem to be realistic.**
> > >
> > > **RESPONSE:**
> > > * In fact, our experiment setting that uses the original MuJoCo environment as the target domain enables direct comparison with any offline RL papers that use D4RL datasets for evaluation, as the benchmark dataset and environment are the same and the scores are comparable.
> > > * Regarding reporting the unnormalize scores. Actually, conversion between normalized and unnormalized scores is simple and only involves simple math ($\text{normalized score}=100*(\text{score}-\text{random score})/(\text{expert score}-\text{random score})$, where $\text{random score}$ and $\text{expert score}$ for each task can be found in the D4RL paper). The D4RL paper reports benchmark results with both normalized and unnormalized scores. If the reviewer thinks it is necessary, we are happy to convert the scores in the paper to normalized scores.
> > >
> > > > **"Moreover, the design of source tasks doesn't seem to be realistic."**
> > >
> > > **RESPONSE:**
> > > Using a simulator with a large dynamics gap will lead to a significantly harder sim-to-real problem as compared to the cases with simulators that only have mild differences with the target domain. If the dynamics gap is small, then the necessity of sim-to-real transfer is less important, even learning an RL policy entirely in simulation and performing a hard transfer is likely to work. On the hand, when the dynamics gap is large, then only models with strong sim-to-real transfer capabilities are likely to solve the tasks. As a result, we choose the more challenging setting, and designed a variety of perturbed dynamics as well as real-world experiments to evaluate our model. We believe this can better demonstrate the capability of our approach.
> > >
> > > > **"I agree that the experiments with a real-world wheel-legged robot are a great result, but the literature review of sim-to-real works is limited (in section 2.1), and no baselines in the experiment (I think DARC and DARA are not sim-to-real work)."**
> > >
> > > **RESPONSE:** Both DARC and DARA address sim-to-real transfer issues using the RL framework between the source (i.e. a simulator) and target (i.e. the real world) domains. We quote the supporting paragraphs in their original papers as follows:
> > >
> > > * **DARC:** "In this paper, we examine a specific transfer learning scenario that we call domain adaptation, by analogy to domain adaptation problems in computer vision (Csurka, 2017), where the training process in a source domain can be modified so that the resulting policy is effective in a given target domain."
> > > * **DARA:** "Specifically, DARA emphasizes learning from those source transition pairs that are adaptive for the target environment and mitigates the offline dynamics shift by characterizing state-action-next-state pairs instead of the typical state-action distribution sketched by prior offline RL methods."
> > >
> > > ## Response to "Re:(7)"
> > > > **"Maybe the scale of reward between Appendix B.3 and Figure 4 should be aligned."**
> > >
> > > **RESPONSE:** In Figure 4, the two sub-figures are for two different tasks, which have different reward functions. So it is not very meaningful to align their scale of reward.

---

> > > ### Author Response · Authors · 2022-08-08
> > > **Response to the follow-up questions (1/3)**
> > >
> > > We thank the reviewers for the detailed feedbacks and advices that potentially help to improve our work. Further explanations and additional experiments are presented below.
> > >
> > > ## Response to "Re:(1)"
> > > > **"whether "Hybrid Offline and Online" describes what this paper address well. For example, [1,2,3] can be called a "Hybrid Offline and Online" RL even though they tackle single dynamics since they leverage offline demonstration to accelerate online training. The key of this paper seems to be "dynamics-awareness", "off-dynamics", or "sim-to-real", or else. "**
> > >
> > > **RESPONSE:**
> > > * Yes, we focus on handling the dynamics gap and trying to merge offline and online RL for sim-to-real settings. Actually, as indicated in the title, abstract, and introduction of the paper, we term our model "Dynamics-Aware Hybrid Offline-and-Online RL".
> > > * The reference [1,2,3] mentioned by the reviewer also have some distinctions with our work, as they incorporate offline demonstrations by imitation learning and serve as an auxiliary design in the RL training process, rather than integrating both offline RL and online RL in a single framework.
> > >
> > >
> > > ## Response to "Re:(2)"
> > > > **"If these analyses are easily applicable to continuous control, these statements should be included in the main text explicitly (or Appendix). I guess the tabular case analysis also works well in a practical setting in some cases, but the rigorous analysis may have some obstacles."**
> > >
> > > **RESPONSE:** Thanks for your comment. All our simulation and real-world experiments are continuous control tasks, and our empirical results show that H2O indeed works well on these tasks. In a similar vein, the theoretical analysis of other value regularization methods like CQL, COMBO, and PessORL are all conducted on tabular settings, and they also show good performance on continuous control tasks. We will include the statement and corresponding discussions in the final version of the paper to further clarify this.
> > >
> > > ## Response to "Re:(3)"
> > > > **"I appreciated the authors for clarification about the relationship among previous offline works (CQL, COMBO, PessORL, and CPQ) and off-dynamics works (DARC, DARA). Since H2O also could be considered as having the reward corrective term to handle the dynamics gap (Eq. 11), I think it would be helpful to include short note about how they are different in the related work section."**
> > >
> > > **RESPONSE:** The reward corrective term in H2O is mainly for theoretical analysis to illustrate how H2O works. It is implicit and does not appear in our practical algorithm. However, we would like to thank the reviewer for the suggestion. We attempted to add discussions in the related work section, but it resulted in exceeding the page limit. We will definitely consider including the discussion in the final version of the paper if there is more space.
> > >
> > > > **"I think this should be carefully stated, since the collected-demonstration quality in the real-world experiment seems high (Fig 3). Whether H2O does not suffer from low-quality real-world data does not seem to be evaluated."**
> > >
> > > **RESPONSE:**
> > > * To further verify the performance of H2O vs DARC-style methods on tasks with low-quality offline datasets, we have conducted additional experiments on Halfcheetah-Friction task with the random offline dataset in D4RL. The random dataset is generated by a random policy and the quality can be quite low. Due to limited time during rebuttal, we report the scores with 3 seeds in the following table:
> > >
> > > |Dataset| Unreal Dynamics|SAC|CQL|DARC|DARC+|H2O|
> > > |:-----:|:-----:|:-----:|:-----:|:-----:|:-----:|:-----:|
> > > |HalfCheetah_Random|Friction|2684±2646|2465±180|537±250|425±99|**4862±1608**|
> > >
> > > * It can be observed that DARC-style algorithms indeed perform badly when given low-quality data, which is due to their theoretical foundation of trajectory distribution divergence minimization. Again, we find H2O performs very well even given the random dataset, which greatly surpasses the performance of pure online or offline baselines.

---

### Official Review · Reviewer_eEjG · 2022-07-09

**Rating:** 6
**Confidence:** 3
**Soundness:** 3 good
**Presentation:** 2 fair
**Contribution:** 3 good

**Summary:**

This paper presents a method for learning from a combination of offline data collected for example in simulation where the dynamics may not necessarily be accurate, as well as data from the real-world with the “true” dynamics. The method is built off of the CQL offline RL algorithm and proposes a few techniques for incorporating additional offline data, including adaptive value regularization based on estimated dynamics gaps from learned classifiers, and reweighting the bellman error based on the shifting dynamics. The method is evaluated on simulated MuJoCo environments with modified dynamics, as well as a real world experiment using a wheeled robot to perform a balancing task.


**Questions:**

- It is surprising to me that the performance of H2O-a is very similar to that of H2O. My understanding is that H2O replaces the learned $\omega(s, a)$ with a fixed, uniform $\omega(s, a)$. But, doesn’t this remove the capability for the algorithm to detect any off-dynamics transitions at all? Doesn’t the dynamics ratio to fix the Bellman error “-dr” also depend on the learned $\omega$? If so, why doesn’t it have the same performance as H2O-a-dr? Some clarification would be appreciated.

Nitpicks/typos:
- L128: the notation for the behavior policy is introduced but it doesn’t seem like it’s actually used in equation 3?
- L141: sate-action -> state-action
- L165: the way the maximization term is written is not consistent with Eq. 4
- L173: close-form -> closed-form
- Fig 2: epoches -> epochs

**Limitations:**

The limitations of the work are not described in detail in the last section, although the authors provide a few potential future directions. I would encourage the authors to more explicitly address the limitations of the current work as well as potential negative societal impacts.


**Strengths And Weaknesses:**

Strengths:
- The problem of learning from real, offline data as well as a potentially inaccurate simulator is important and applicable, and to my knowledge, relatively unexplored. This work is a nice step towards being able to learn in this setting.
- The formulation of the H2O objective is well-motivated and has a useful interpretation in terms of “confidences” in each state-action pair, depending on whether they are in-distribution for the offline data or have high dynamics gaps. The analysis in section 5 is quite nice.
- The experimental results are strong, and particularly the real-world wheeled robot experiment helps to convince me that this method is able to handle in-the-wild situations where we have access to some real offline data and a simulator.

Weaknesses:
- Mostly, I am confused about the conclusions drawn from the ablation study. Based on the ablation study, it seems that the performance improvements of H2O over CQL in the simulation experiments are largely due to the reweighting of the Bellman updates rather than the value regularization part of the framework. I disagree with the statement that “Removing the entire value regularization part from H2O leads to substantial performance degradation”, as the performance of H2O vs H2O-reg in the table is within the standard error range.
- The overall clarity of the writing could be improved. Below I note some specific points but I think it could be helpful to have some additional proofreading.

Clarity:
- I think the clarity of section 6.1, detailing the experimental setups in simulation, could be improved. Specifically, I think it would help to clarify that the original MuJoCo environments are being considered the “real” environments, and the offline dataset described in L264 serves as offline data from the real world.
- I don’t quite understand from reading the main paper how the objective derived in Section 4 is incorporated into the main algorithm. The algorithm box in appendix B.1 helps significantly, but I feel it would help to mention that this objective is being used during online learning.
- The introduction of $\omega$ in section 4.2 is rather confusing – it’s not described how $\omega$ characterizes the dynamics gaps for the samples until later. I might recommend providing the KL divergence definition of $\omega$ earlier or reformatting the writing slightly.

---

> ### Author Response · Authors · 2022-08-02
> **Response to Reviewer eEjG (2/2)**
>
> ## Regarding the clarity of the paper
>
> > **Weakness 2: "The overall clarity of the writing could be improved. Below I note some specific points but I think it could be helpful to have some additional proofreading."**
>
> > **Clarity 1: "I think it would help to clarify that the original MuJoCo environments are being considered the “real” environments, and the offline dataset described in L264 serves as offline data from the real world"**
>
> > **Clarity 2: "I don’t quite understand from reading the main paper how the objective derived in Section 4 is incorporated into the main algorithm. The algorithm box in appendix B.1 helps significantly, but I feel it would help to mention that this objective is being used during online learning."**
>
> > **Clarity 3: "The introduction of $\omega$ in section 4.2 is rather confusing – it’s not described how  $\omega$ characterizes the dynamics gaps for the samples until later. I might recommend providing the KL divergence definition of $\omega$ earlier or reformatting the writing slightly."**
>
> **RESPONSE:** We appreciate the reviewer for the constructive suggestion regarding the presentation of this paper. We have revised the following part of the paper to address the concerns of the reviewer:
>
> * We have revised a number of typos and inaccurate descriptions in the updated version of the paper.
> * We have added an additional description of the source and target environment in Appendix B.3 in the updated version of the paper.
> * In our original version of Section 4.2, we want to first present to the readers the whole picture of our framework, and then describe the detailed design of each component. Hence we first introduce $\omega(\mathbf{s}, \mathbf{a})$ without describing its details, and then introduce $u(\mathbf{s}, \mathbf{a})$ and how it is used to compute $\omega$. If the reviewer thinks that it will cause unnecessary confusion, we are happy to revise it in the final version of the paper.
>
> ## Other comments:
>
> > **"L128: the notation for the behavior policy is introduced but it doesn’t seem like it’s actually used in equation 3?"**
>
> **RESPONSE:** We thank the reviewer for pointing this out. $\pi_\beta$ (the latest version has changed it into $\pi_\mathcal{D}$ according to the suggestion from another reviewer) is actually implicitly captured by sampling state-action pairs from dataset $\mathcal{D}$ (i.e., $(\mathbf{s},\mathbf{a})\sim \mathcal{D}$ in Eq. (3)). It is equivalent to write it as $\mathbf{s}\sim \mathcal{D},\mathbf{a}\sim \pi_\beta$.
>
> > **"L165: the way the maximization term is written is not consistent with Eq. 4"**
>
> **RESPONSE:** We thank the reviewer for pointing out this notational inconsistency. We have revised it in the updated version.

---

> ### Author Response · Authors · 2022-08-02
> **Response to Reviewer eEjG (1/2)**
>
> We thank the reviewer for the thorough and detailed comments. Regarding the concerns from the reviewer, we describe the detailed response as follows:
>
> ## Regarding the ablation study
> > **Weakness 1: "Based on the ablation study, it seems that the performance improvements of H2O over CQL in the simulation experiments are largely due to the reweighing of the Bellman updates rather than the value regularization part of the framework. I disagree with the statement that “Removing the entire value regularization part from H2O leads to substantial performance degradation”, as the performance of H2O vs H2O-reg in the table is within the standard error range."**
>
> > **Question: "It is surprising to me that the performance of H2O-a is very similar to that of H2O. My understanding is that H2O replaces the learned $\omega(\mathbf{s},\mathbf{a})$ with a fixed, uniform $\omega(\mathbf{s},\mathbf{a})$. But, doesn’t this remove the capability for the algorithm to detect any off-dynamics transitions at all? Doesn’t the dynamics ratio to fix the Bellman error “-dr” also depend on the learned ω? If so, why doesn’t it have the same performance as H2O-a-dr?"**
>
> **RESPONSE:** The results reported in Table 2 were averaged over 3 random seeds. We realized that the reported score of H2O is actually lower than our 5-seed result in Table 1. We have updated the results of H2O, CQL and SAC in Table 2 to be consistent with Table 1 in our revised paper. Due to the limited rebuttal period for additional experiments, we will provide the updated scores for all variants of H2O using 5 random seeds during the discussion period.
>
> * Regarding the issue that H2O, H2O-a and H2O-reg have similar performance:
>     * We want to clarify that the same dynamics ratio information $P_{\widehat{\mathcal{M}}}/P_\mathcal{M}$ are used in both computing the adaptive weight $\omega(\mathbf{s},\mathbf{a})$ as well as the importance-sampling weights $P_\mathcal{M}/P_{\widehat{\mathcal{M}}}$ to fix the Bellman error. Hence even without using the adaptive regularization weights $\omega(\mathbf{s},\mathbf{a})$, similar dynamics adaptive information is applied when fixing the Bellman error.
>     * Based on the updated 5-seed result of H2O, we observe H2O indeed improves over H2O-a and H2O-reg (H2O: 6813±289 vs H2O-a: 6568±194 vs H2O-reg: 6396±175). Moreover, including the adaptive weights $\omega$ is a good bargain, since it relies on the same dynamics ratio $P_{\widehat{\mathcal{M}}}/P_\mathcal{M}$ needed in the importance-sampling weights, which does not add much computation burden but leads to improved performance.
> * Regarding using uniform $\omega$ removes the capability for detecting off-dynamics transitions:
>     * The answer is no, as the importance-sampling weights $P_\mathcal{M}/P_{\widehat{\mathcal{M}}}$ also carries the information to distinguish off-dynamics transitions. For example, if the simulation environment has a high probability to make the transition from $(\mathbf{s}, \mathbf{a}) \rightarrow \mathbf{s}^{\prime}$ ($P_{\widehat{\mathcal{M}}}(\mathbf{s},\mathbf{a}, \mathbf{s}^{\prime})$ is large), but it rarely happens in the real environment ($P_{\mathcal{M}}(\mathbf{s},\mathbf{a}, \mathbf{s}^{\prime})$ is small), then the importance-sampling weight will be small and guide the Q-function to ignore learning from this transition.
> * Regarding the relationship with "-dr" and "-a":
>     * We apologize for the confusion. In our ablation study, "-dr" refers to removing the importance-sampling weights $P_\mathcal{M}/P_{\widehat{\mathcal{M}}}$ used in fixing the Bellman error. Both $\omega$ and importance-sampling weights $P_\mathcal{M}/P_{\widehat{\mathcal{M}}}$ depends on the dynamics ratio $P_{\widehat{\mathcal{M}}}/P_\mathcal{M}$ evaluated using the two discriminators. Given the estimated dynamics ratio, the importance-sampling weight $P_\mathcal{M}/P_{\widehat{\mathcal{M}}}$ and $\omega$ are implemented independently in our ablation study. Finally, H2O-a-dr refers to the variants that use uniform value regularization without Bellman error correction, which is different from H2O-dr which still uses adaptive weight $\omega$ for value regularization.

---

> > ### Comment · Reviewer_eEjG · 2022-08-05
> > **Thank you for the response**
> >
> > Thank you for your detailed response to my review. I appreciate the clarification about the ablation experiments and hope that the authors can make it a bit more clear in the exposition also for the different ablation settings. The additional simulated experiments for walker also strengthen the case for this work. I continue to vote for acceptance particularly because I believe that the real world experiments are significant.

---

> > > ### Author Response · Authors · 2022-08-08
> > > **Thank you for the response!**
> > >
> > > We really appreciate your comments and would keep improving the work!

---

### Official Review · Reviewer_hCpT · 2022-07-11

**Rating:** 6
**Confidence:** 3
**Soundness:** 3 good
**Presentation:** 3 good
**Contribution:** 3 good

**Summary:**

This paper is based on the premise that we have an imperfect simulator for online RL, as well as a fixed dataset for offline RL. It proposes a new policy evaluation objective that is designed to penalize the Q values at (s,a) pairs where there is a big gap between simulation and real. The gap is measured by two discriminators trained following DARC.

With manually modified environment dynamics on MuJoco-HalfCheetah task as well as a real robot experiment, the authors showed that their algorithm is able to better utilize both the imperfect simulator and the limited but accurate offline data.


**Questions:**

* (Apologies if those details are already mentioned in the appendix.)
* What is the distribution of the human performance on the collected dataset for the real robot?
* For l173, the exact closed form solution I got is d = w(s,a) exp (Q(s,a) - 1). Can you double check if that is correct, and comment on whether it is preferable to use the exact solution instead of the proportional one in l173?
* In l184-185, what is the ratio r of experience from source vs. target that you used for the DARC like training? And did you try to tune that value?

Comments:
* In Table 2, it is interesting that H2O-a is not that bad, indicating that perhaps the environment deviation is not really dependent on (s,a). It’d be interesting to look at environments where the sim-real gap is uneven across (s,a) pairs, e.g. the dead-zones mentioned in l282-283.

typos/nits:
* L55: incorporating -> incorporate
* L56: only restricted -> are only restricted
* l296 : which does not impacted -> which is not impacted

**Limitations:**

The requirement of training two discriminators constrains the proposed method’s application to domains where enough offline dataset is available for training a discriminator.

The approximation of the expected value in l207 can be undefined for s’ unvisited in the offline dataset.


**Strengths And Weaknesses:**

Strengths:
* The paper is well written, especially the introduction and section 4, with clear steps and explicitly listed assumptions.
* The authors did experiments both in the real and the simulated environments, validating their main claim that this method helps bridge the sim2real gap and makes RL more applicable in the real world.
* The method proposed is original as far as the reviewer can tell.

Weaknesses:
* Some parts of the paper can benefit from a bit more clarification. See the Questions section below.

---

> ### Author Response · Authors · 2022-08-02
> **Response to Reviewer hCpT (2/2)**
>
> > **(3) "In l184-185, what is the ratio r of experience from source vs. target that you used for the DARC like training? And did you try to tune that value?"**
>
> **RESPONSE:** We thank the reviewer for the question. The original version of DARC leverages online data collected every r steps (periodically collecting fresh rollout data from the target environment), which is incompatible with our setting as we do not allow real environment interaction during policy training. In order to have a fair comparison with other methods, we remove the extra real environment data collection as in DARC, and only use the fixed offline target dataset for training. When training DARC in our experiments, we maintain a fixed size buffer for simulation data, which is 10 times that of the offline dataset. We have described the difference in DARC implementation in Appendix C.4 in our paper.
>
> ## Regarding the Comments:
> > **(4) "In Table 2, it is interesting that H2O-a is not that bad, indicating that perhaps the environment deviation is not really dependent on (s, a). It’d be interesting to look at environments where the sim-real gap is uneven across (s,a) pairs, e.g. the dead-zones mentioned in l282-283."**
>
> **RESPONSE:** The results reported in Table 2 were averaged over 3 random seeds. We realized that the reported score of H2O is actually lower than our 5-seed result in Table 1. We have updated the results of H2O, CQL, and SAC in Table 2 to be consistent with Table 1 in our revised paper. Due to the limited rebuttal period for additional experiments, we will provide the updated scores for all variants of H2O using 5 random seeds during the discussion period.
>
> Regarding the issue that H2O and H2O-a have similar performance:
> * We want to clarify that the same dynamics ratio information $P_{\widehat{\mathcal{M}}}/P_\mathcal{M}$ are used in both computing the adaptive weight $\omega(\mathbf{s},\mathbf{a})$ as well as the importance-sampling weights $P_\mathcal{M}/P_{\widehat{\mathcal{M}}}$ to fix the Bellman error. Hence even without using the adaptive regularization weights $\omega(\mathbf{s},\mathbf{a})$, similar dynamics adaptive information is applied when fixing the Bellman error.
> * Based on the updated 5-seed result of H2O, we observe H2O indeed improves over H2O-a (H2O: 6813±289 vs H2O-a: 6675±179). Moreover, including the adaptive weights $\omega$ is a good bargain, since it relies on the same dynamics ratio information $P_{\widehat{\mathcal{M}}}/P_\mathcal{M}$ needed in the importance-sampling weights, which does not add much computation burden but leads to improved performance.
>
> ## Regarding the Limitations:
> > **(5) "The requirement of training two discriminators constrains the proposed method’s application to domains where enough offline dataset is available for training a discriminator."**
>
> **RESPONSE:** In our problem setting, we assume the availability of the offline dataset of the target domain, which is used to facilitate correcting the dynamics gap during online learning. Note that H2O adopts a different treatment as in DARC, the offline data is not only used for training discriminators, but also used in policy evaluation. As the discriminators are trained in a supervised manner, which in principle has higher data efficiency than RL. Moreover, with the complement of online simulation data, H2O can be less data-demanding compared with pure offline RL approaches.
>
> > **(6) "The approximation of the expected value in l207 can be undefined for s’ unvisited in the offline dataset."**
>
> **RESPONSE:**
> We thank the reviewer for this comment. In fact, the transition dynamics $P_\mathcal{\widehat{M}}$ and $P_\mathcal{M}$ are properties associated with the environments rather than the collected dataset. In H2O, the dynamics ratio $P_\mathcal{\widehat{M}}/P_\mathcal{M}$ is evaluated using a pair of discriminators in a sample-based manner rather than directly approximating the two dynamics distributions and computing their ratio. The approximation $P_{\widehat{\mathcal{M}}}=N(s^{\prime},\hat{\Sigma}_{\mathcal{D}})$ is only used to sample candidate next states $\hat{\mathbf{s}}^{\prime}$ based on triplets $(\mathbf{s}, \mathbf{a}, \mathbf{s}^{\prime})$ from the simulated replay buffer $B$. Thus, $\hat{\mathbf{s}}^{\prime}$ that are unvisited in offline dataset won't be an issue for the computation of $u(\mathbf{s},\mathbf{a})$ in our implementation. And we do observe the dynamics gap measure $u(\mathbf{s},\mathbf{a})$ computed in this way is much more stable and generalizes better on unseen $(s, a, s^{\prime})$ triples compared with directly approximate the dynamics distributions $P_\mathcal{\widehat{M}}$ and $P_\mathcal{M}$ and compute their KL divergence.

---

> > ### Comment · Reviewer_hCpT · 2022-08-07
> > **Response to authors' comments**
> >
> > Thank you authors for your detailed replies to my questions. Here are some follow up comments:
> >
> > ## Regarding the comment to (2)
> > In your response, you used the assumption “If we assume the dynamics gap distribution w(s,a) > 0 holds for all state-action pairs”. That is certainly a reasonable assumption, but I’m wondering if it would be simpler if in eq(5), you state d^theta(s,a) > 0 instead of >= 0.
> > The rest of the derivation makes sense. Thank you for writing it out. It’s clear to me now.
> >
> >
> > ## Regarding the comment to (4)
> > Thank you for the response. I overlooked the fact that the dynamics ratio is used for the importance sampling weights as well. That makes sense.
> >
> >
> > ## Regarding the comment (5) (6)
> > I understand your approximation to the expected value of s’. I was suggesting that for unvisited (s,a) or (s,a,s’), the dynamics ratio in eq(7) can have unstable values. I see that in B.2 “discriminators” section, you deployed several tricks to help discriminators be stable. E.g. Without the 2xTanh in l684, the ratio (1-D) / D could be unstable in eq 28. In addition, the gaussian noise added to the input dimensions are domain specific. In short, I agree that supervised methods are in principle more data efficient than RL, but discriminators are quite tricky to train in practice. That’s why I treat their usage as a limitation.

---

> > > ### Author Response · Authors · 2022-08-08
> > > **Thank you for the response!**
> > >
> > > We really appreciate your positive feedbacks:
> > >
> > > > **Response to "Regarding the comment to (2)"**
> > >
> > > **RESPONSE:** We agree with your comment and would take it into consideration for our final version.
> > >
> > > > **Response to "Regarding the comment to (5)(6)"**
> > >
> > > **RESPONSE:** We agree with the comment that the discriminators need special designs for stable predictions. Therefore, as we mentioned in the conclusion section in the paper, we would always bear this in mind as a future direction until developing some more decent and practical implementations on dynamics gap quantification.

---

> ### Author Response · Authors · 2022-08-02
> **Response to Reviewer hCpT (1/2)**
>
> We sincerely appreciate the reviewer for the constructive comment and the positive feedback on our work. Regarding the concerns from the reviewer, we describe the detailed response as follows:
>
> ## Regarding the Questions:
> >**(1) "What is the distribution of the human performance on the collected dataset for the real robot?"**
>
> **RESPONSE:** We thank the reviewer for the comment. We have added the per-step reward distribution of the collected datasets in wheel-legged robot tasks in Appendix B.4.
>
> >**(2) "For l173, the exact closed form solution I got is d = w(s,a) exp (Q(s,a) - 1). Can you double check if that is correct, and comment on whether it is preferable to use the exact solution instead of the proportional one in l173?"**
>
> **RESPONSE:** We thank the reviewer for reminding us. In the following, we provide the complete derivation of Eq.(6) in our paper.
>
> The Lagrangian of the primal optimization problem in Eq. (5) is given by:
>
>   $\mathcal{L}\left(d^{\phi}; \mu, \boldsymbol{\lambda}\right)=E_{\mathbf{s},\mathbf{a} \sim d^{\phi}} Q(\mathbf{s},\mathbf{a})-D_{KL}\left(d^{\phi}(\mathbf{s},\mathbf{a}) \| \omega(\mathbf{s},\mathbf{a})\right)+\mu\left(\sum_{\mathbf{s},\mathbf{a}} d ^{\phi}(\mathbf{s},\mathbf{a})-1\right)+\boldsymbol{\lambda}(\mathbf{s},\mathbf{a}) d^{\phi}(\mathbf{s},\mathbf{a})$
>
>   where $\mu$ is the Lagrange dual variable for the constraint $\sum_{\mathbf{s},\mathbf{a}} d ^{\phi}(\mathbf{s},\mathbf{a})=1$, and $\boldsymbol{\lambda}(\mathbf{s},\mathbf{a})$ is the Lagrangian dual variable for positivity constraints on $d^{\phi}$. Setting the gradient of the Lagrangian w.r.t. $d^\phi$ to 0 yields:
>
>   $d^{\phi *}\left(\mathbf{s},\mathbf{a}\right) =\omega(\mathbf{s},\mathbf{a}) \exp [Q(\mathbf{s},\mathbf{a})+\mu+\boldsymbol{\lambda}(\mathbf{s},\mathbf{a})-\mathbf{1}], \forall (\mathbf{s},\mathbf{a})\in \mathcal{S}\times\mathcal{A}$
>
>   If we assume the dynamics gap distribution $\omega(s, a)> 0$ holds for all state-action pairs, hence $d^{\phi *}\left(\mathbf{s},\mathbf{a}\right)> 0$ trivially holds, which implies $\boldsymbol{\lambda}(\mathbf{s}, \mathbf{a})=0$ for each state-action pair according to the complementary slackness condition.
>   Utilizing the normalization constraint $\sum_{\mathbf{s},\mathbf{a}} d ^{\phi}(\mathbf{s},\mathbf{a})=1$, we have:
>
>   $\sum_{\mathbf{s},\mathbf{a}} d^{\phi *}(\mathbf{s},\mathbf{a}) =\sum_{\mathbf{s},\mathbf{a}} \omega\left(\mathbf{s},\mathbf{a}\right) \exp [Q(\mathbf{s},\mathbf{a})] e^{\mu-\mathbf{1}}=1$
>
>   Solving $e^{\mu-1}$ using the above equation and plugging it into the previous equation, we then obtain the final closed-form solution for $d^{\phi *}$:
>
>   $d^{\phi *}\left(\mathbf{s},\mathbf{a}\right)=\frac{\omega(\mathbf{s},\mathbf{a}) \exp [Q(\mathbf{s},\mathbf{a})] }{\sum_{\mathbf{s},\mathbf{a}} \omega(\mathbf{s},\mathbf{a}) \exp [Q(\mathbf{s},\mathbf{a})]}\propto \omega(\mathbf{s},\mathbf{a}) \exp [Q(\mathbf{s},\mathbf{a})]$
>
>   Note that if we plug the exact solution $d^{\phi *}$ and the regularization term $\mathcal{R}(d^{\phi})=-D_{KL}(d^{\phi}||\omega)$ in Eq. (4), we have:
>
>   $\beta\left[E_{\mathbf{s}, \mathbf{a}\sim  d^\phi(\mathbf{s}, \mathbf{a}) }[Q(\mathbf{s}, \mathbf{a})-\log(d^\phi(\mathbf{s}, \mathbf{a})/\omega(\mathbf{s}, \mathbf{a}))]-E_{\mathbf{s},\mathbf{a} \sim \mathcal{D}}[Q(\mathbf{s}, \mathbf{a})] \right]+\widetilde{\mathcal{E}} \left(Q, \hat{\mathcal{B}}^{\pi} \hat{Q}\right)$
>
>   $=\beta\left[E_{\mathbf{s}, \mathbf{a}\sim  d^\phi(\mathbf{s}, \mathbf{a}) }\left[Q(\mathbf{s},\mathbf{a})-\log\left(\frac{\exp [Q(\mathbf{s},\mathbf{a})] }{\sum_{\mathbf{s},\mathbf{a}} \omega(\mathbf{s},\mathbf{a}) \exp [Q(\mathbf{s},\mathbf{a})]} \right) \right]-E_{\mathbf{s},\mathbf{a} \sim \mathcal{D}}[Q(\mathbf{s}, \mathbf{a})] \right]++\widetilde{\mathcal{E}} \left(Q, \hat{\mathcal{B}}^{\pi} \hat{Q}\right)$
>
>   $=\beta\left[E_{\mathbf{s}, \mathbf{a}\sim  d^\phi(\mathbf{s}, \mathbf{a})}\left[
>     \log\left(\sum_{\mathbf{s},\mathbf{a}} \omega(\mathbf{s},\mathbf{a}) \exp [Q(\mathbf{s},\mathbf{a})] \right) \right]-E_{\mathbf{s},\mathbf{a} \sim \mathcal{D}}[Q(\mathbf{s}, \mathbf{a})] \right]+\widetilde{\mathcal{E}} \left(Q, \hat{\mathcal{B}}^{\pi} \hat{Q}\right)$
>
>   $=\beta\left[\log\sum_{\mathbf{s},\mathbf{a}} \omega(\mathbf{s},\mathbf{a}) \exp [Q(\mathbf{s},\mathbf{a})]  -E_{\mathbf{s},\mathbf{a} \sim \mathcal{D}}[Q(\mathbf{s}, \mathbf{a})] \right]+\widetilde{\mathcal{E}} \left(Q, \hat{\mathcal{B}}^{\pi} \hat{Q}\right)$
>
>   In the last step, we can remove $E$ as $\log\sum_{\mathbf{s},\mathbf{a}} \omega(\mathbf{s},\mathbf{a}) \exp [Q(\mathbf{s},\mathbf{a})]$ is a value that does not depend on $(s,a)$ any more. The objective in the last equation is exactly the objective we have used in Eq. (6) in the paper. Note that the derivation is based on the exact solution of $d^{\phi *}$ rather than the proportional one.

---

### Author Response · Authors · 2022-08-02
**Revision Summary**

We would like to thank the reviewers for their detailed comments. We have updated the paper to address the concerns and suggestions from the reviewers. Summary of updates are as follows:

1. We conducted additional experiments on Walker2d Medium-Replay tasks to further verify our method in simulation environments. Experiment results are presented in Appendix C.2.
2. We added additional results on a variant of H2O that uses reverse KL to calculate the dynamics gap measure $u(\mathbf{s},\mathbf{a})$. Detailed results and discussion are presented in Appendix C.3.
3. We added additional clarification and discussion of the source and target domain in our simulation-based experiments in Appendix B.3.
4. We reported the per-step reward distribution of the collected datasets in real-world wheel-legged robot tasks in Appendix B.4 Figure 3. The plots of the cumulative reward of H2O and baselines in real-world validation are presented in Appendix B.4 Figure 4.
5. We updated the results of H2O, CQL, and SAC in Table 2 with 5 random seeds (the original version uses 3 random seeds).
6. We have fixed all the typos, confusing mathematical expressions, and inaccurate reference information as pointed out by reviewers.

---

> ### Author Response · Authors · 2022-08-06
> **Additional revisions**
>
> Dear reviewers, we have updated all the ablation results of H2O in Table 2 using 5 random seeds (also revised in the latest version of the paper). The results of Table 2 reported in the initial submission were averaged over 3 random seeds. For your convenience, we present the changes in scores as follows:
>
> |HalfCheetah_Gravity|H2O|H2O-a|H2O-dr|H2O-a-dr|H2O-reg|H2O-reg-dr|CQL|SAC
> |:-------------------|:---:|:---------:|:---------:|:---------:|:---------:|:----------:|:---------:|:---------:|
> |Average on 3 seeds (initial) |6642±324|6568±194|4866±303|5381±406|6396±175|5322±345|5947±59|5001±548|
> |Average on 5 seeds (updated) |6813±289|6675±179|4721±196|5223±198|6501±147|5290±356|5774±214|4513±513|

---

### Meta-Review · Area_Chair_ZCZa · 2022-08-26

**Recommendation:** Accept
**Confidence:** Certain

**Metareview:**

The authors present a novel but realistic problem, where you want to learn from limited offline real-world data, and from unlimited simulation data, as is generally the case for sim2real with real-world finetuning. Experiments are performed on D4RL HalfCheetah with three artificially created dynamics gaps, as well as real-world wheeled robot.

Thanks to the reviewers and authors for engaging in active discussions in experimental details, and appreciate the authors for updating the draft with additional experiments with TD3+BC, Random_HalfCheetah etc. I recommend the paper for acceptance, as this paper could encourage further research into this important problem setting.


**Award:**

No

---

### Decision · Program_Chairs · 2022-09-14

Accept